# Chlorine bridge bond-enabled binuclear copper complex for electrocatalyzing lithium−sulfur reactions

Qin Yang[1,7], Jinyan Cai[2,7], Guanwu Li[3], Runhua Gao[4], Zhiyuan Han[4], Jingjing Han[5], Dong Liu[5], Lixian Song[1], Zixiong Shi[6], Dong Wang[3], Gongming Wang ®[2] ✉, Weitao Zheng ®[3], Guangmin Zhou ®[4] ✉ & Yingze Song ®[1] ✉

Engineering atom-scale sites are crucial to the mitigation of polysulfide shuttle, promotion of sulfur redox, and regulation of lithium deposition in lithium−sulfur batteries. Herein, a homonuclear copper dual-atom catalyst with a proximal distance of 3.5 Å is developed for lithium−sulfur batteries, wherein two adjacent copper atoms are linked by a pair of symmetrical chlorine bridge bonds. Benefiting from the proximal copper atoms and their unique coordination, the copper dual-atom catalyst with the increased active interface concentration synchronously guide the evolutions of sulfur and lithium species. Such a delicate design breaks through the activity limitation of mononuclear metal center and represents a catalyst concept for lithium−sulfur battery realm. Therefore, a remarkable areal capacity of 7.8 mA h cm$^{-2}$ is achieved under the scenario of sulfur content of 60 *wt.*%, mass loading of 7.7 mg cm$^{-2}$ and electrolyte dosage of 4.8 μL mg$^{-1}$.

The earth-abundant and eco-friendly sulfur presents a high theoretical specific capacity of 1675 mA h g$^{-1}$, which in turn enables an ultrahigh theoretical energy density of 2600 Wh kg$^{-1}$ for lithium−sulfur batteries (LSBs)[1–3]. In this sense, the rechargeable LSBs are considered as one of the most promising candidates for the next-generation renewable energy storage systems, especially in the context of carbon neutral goal[4,5]. Nevertheless, a multitude of obstacles have severely hindered their commercial application[6,7]. To begin with, the soluble lithium polysulfides (LiPSs) usually diffuse and shuttle between the cathode and anode, resulting in irreversible capacity fading[8–10]. Another detrimental concern lies in the sluggish redox reactions, which accordingly triggers LiPS accumulation and deteriorate sulfur utilization[11,12]. Beyond that, the uncontrolled growth of lithium dendrites on the

anode side also gives rise to the battery performance degeneration and potential safety issues, which further worsen the energy density and duration of LSBs[13,14].

Recently, various electrocatalytic strategies (metal oxides[15,16], sulfides[17], selenides[18,19], phosphides[20,21] and nitrides[22]) have been proposed to combat the aforementioned issues, mainly pertaining to vacancy defects[23–25], heteroatom doping[26–28], and interface engineering[29–31]. Thereof, the introduction of over-weighted metal compounds inevitably degrades the overall energy density of the LSBs[32,33]. To this end, single-atom catalysts (SACs) with atomically dispersed sites anchored on suitable substrates can enable the maximum atomic utilization and robust catalytic activity[34]. To date, multifarious SACs supported by carbon substrates have emerged to

[1]State Key Laboratory of Environment-Friendly Energy Materials, School of Materials and Chemistry, Tianfu Institute of Research and Innovation, Southwest University of Science and Technology, Mianyang 621010, China. [2]Department of Chemistry, University of Science and Technology of China, Hefei 230026, China. [3]Key Laboratory of Automobile Materials MOE, School of Materials Science & Engineering, Jilin Provincial International Cooperation Key Laboratory of High-Efficiency Clean Energy Materials, Jilin University, Changchun 130012, China. [4]Tsinghua-Berkeley Shenzhen Institute & Tsinghua Shenzhen International Graduate School, Tsinghua University Shenzhen, Shenzhen 518055, China. [5]Key Laboratory of Neutron Physics and Institute of Nuclear Physics and Chemistry, China Academy of Engineering Physics, Mianyang 621999, China. [6]Materials Science and Engineering, Physical Science and Engineering Division, King Abdullah University of Science and Technology (KAUST), Thuwal 23955-6900, Saudi Arabia. [7]These authors contributed equally: Qin Yang, Jinyan Cai. ✉e-mail: wanggm@ustc.edu.cn; guangminzhou@sz.tsinghua.edu.cn; yzsong@swust.edu.cn

enhance the electrocatalytic effect and promote the redox kinetics of sulfur cathodes[35–37]. In terms of the lithium anode, SACs also can afford lithiophilic sites to homogenize the lithium nucleation and thus suppress the growth of lithium dendrites[38,39]. Despite these great breakthroughs in LSBs, it remains a tremendous challenge to achieve high-efficiency, stable, and high-loaded SACs[40], e.g., atomic termination[41], dual-atom active metal sites[42,43], and so on. Especially, for dual-atom approach, each active site can contain a pair atom for participating the adsorption or catalytic processes. For instance, the oxygen-bridged indium-nickel atomic pairs recently have been reported as dual-atom active sites to promote $CO_2$ reduction[44]. However, heterogeneous molecule catalysts with atomic metal sites have been scarcely reported in LSB field to date[45]. Moreover, the function of binuclear metal centers in molecule catalysts can be conceived toward the optimal selection for breaking through the activity limitation of mononuclear metal center. Along this line, the dual-metal molecule catalysts are expected to inject new life into the catalyst investigation for LSBs and spark broad interests in the future.

On the proof of this concept, we introduce chlorine (Cl) bridge bond-enabled molecularly dispersed Cu-based complexes as the electrocatalysts in LSBs and investigate the battery performance, involving discharge capacity ($C$), rate capability as well as cycling stability. In detail, the existing Cl bridge bonds lead to the higher activity of homonuclear dual Cu atom catalysts (Cu-2) for the evolutions of sulfur and lithium species than the mononuclear Cu atom catalyst (Cu-1) throughout the whole electrochemical reaction based on the various analysis, including operando Raman spectra, electrochemical detections, synchrotron radiation X-ray three-dimensional nano-computed tomography (X-ray 3D nano-CT), small angle neutron scattering (SANS) and theoretical simulations. Benefiting from the well-designed coordination environment, Cu-2 endows the LSB with a high discharge specific capacity of 1140.6 mA h g$^{-1}$ at 0.2 C and a retention of 91.3% after 100 cycles. Even with a high sulfur loading of 7.7 mg cm$^{-2}$, the S/Cu-2 cathode can obtain an areal capacity ($C_A$) of 7.8 mA h cm$^{-2}$, which propels the real implementation of highly efficient and remarkably durable LSBs.

## Results

The 1-Cuphen and 2-Cuphen complexes are first synthesized by tuning the mole ratio of Cu$^{2+}$ deriving from copper (II) chloride dihydrate and 1,10-phenanthroline (phen) ligand into 1:2 and 1:1, respectively (Fig. 1). The as-obtained Cu complexes (green powder) are loading on the surface of graphene substrate for the synthesizing the Cu-1 and Cu-2 powders (Figs. S1, S2). Note that the large planar aromatic structures of Cu complex molecules can interact with the graphene by the formed π–π interaction. For such a PS$^{2-}$-Li$^+$ coupled LSB system, both of the Cu-2 and Cu-1 can make full use of the active Cu single atoms to effectively regulate Li–S electrochemical reactions. Of particular note, the optimized coordination environment of Cu-2 based on the Cl bridge bonds realizes more remarkable function than Cu-1 in synchronously manipulating the evolutions of sulfur and lithium species. The detailed working mechanism of Cu-2 and Cu-1 are illustrated in Fig. 2. As depicted in Fig. 2a, the Cl bridge bonds give rise to the generation of the homonuclear Cu atoms in pairs for Cu-2. The homonuclear dual Cu centers and their unique coordination with N and Cl atoms endow the Cu-2 with the remarkable advantages in chemically anchoring and kinetically converting LiPSs in contrast to that of Cu-1. Whilst, Cu-2 affords more uniform and lithiophilic nucleation sites to guide the homogeneous Li deposition for effective dendrite suppression than Cu-1 (Fig. 2b).

Ultraviolet−visible (UV–vis) spectra were first collected to confirm the coordination of Cu$^{2+}$ and phen. As displayed in Fig. 3a, the red shift of the characteristic peak located at ~297 nm can be attributed to the formed coordination of Cu$^{2+}$ and phen. Besides, in the visible light region, a broad absorption band (~750 nm) is observed, substantiating the generated Cu$^{2+}$-based complexes[46]. Fig. 3b displays the Fourier transform infrared spectroscopy (FTIR) spectra of phen, 1-Cuphen, and 2-Cuphen. The peaks at 1629 and 1519 cm$^{-1}$ assigning to the stretching vibrations of C=N and C=C bonds in phen aromatic ring respectively shift to 1621 and 1508 cm$^{-1}$ after coordinating with Cu$^{2+}$ in 2-Cuphen. Similar red shift phenomenon can be also observed in the FTIR spectrum of 1-Cuphen. Note that the formation of Cu−N bonds in complexes tends to change the configuration, symmetry and electron

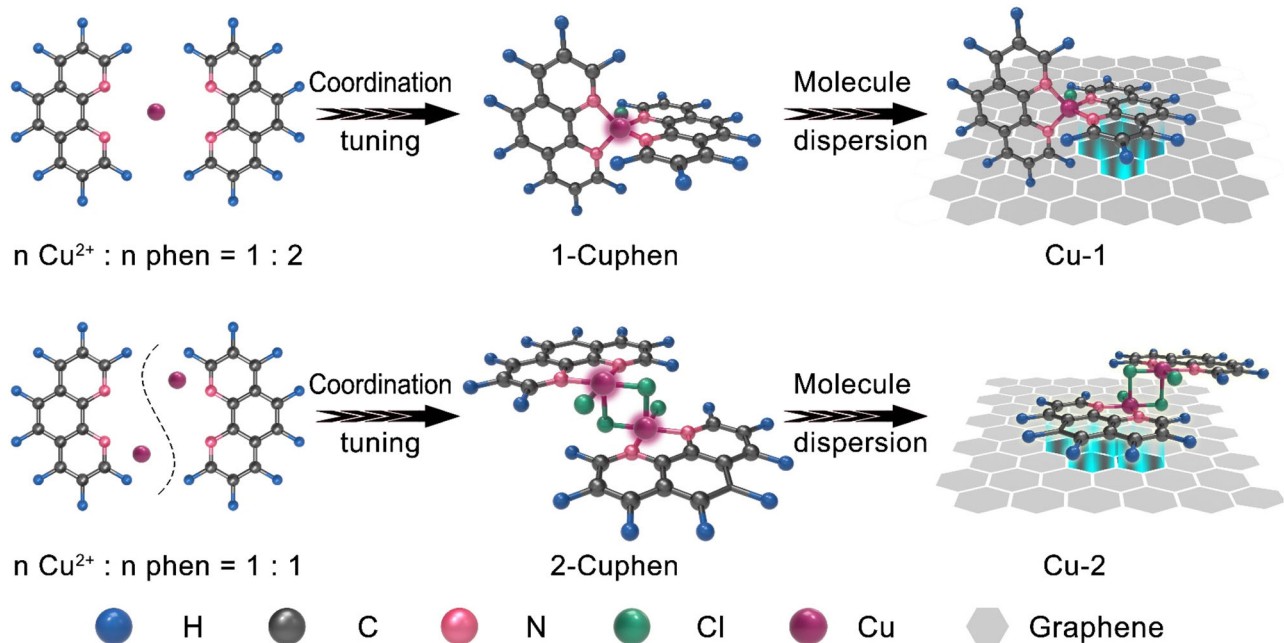

**Fig. 1 | Schematically illustrating the designs of Cu-1 and Cu-2.** Cu-1 is obtained by first tuning the mole ratio of Cu$^{2+}$ and phen as 1:2 to gain 1-Cuphen and then dispersing the 1-Cuphen on the surface of graphene. The Cu-2 is achieved by first adjusting the mole ratio of Cu$^{2+}$ and phen as 1:1 to attain 2-Cuphen, and subsequently dispersing it on the surface of graphene.

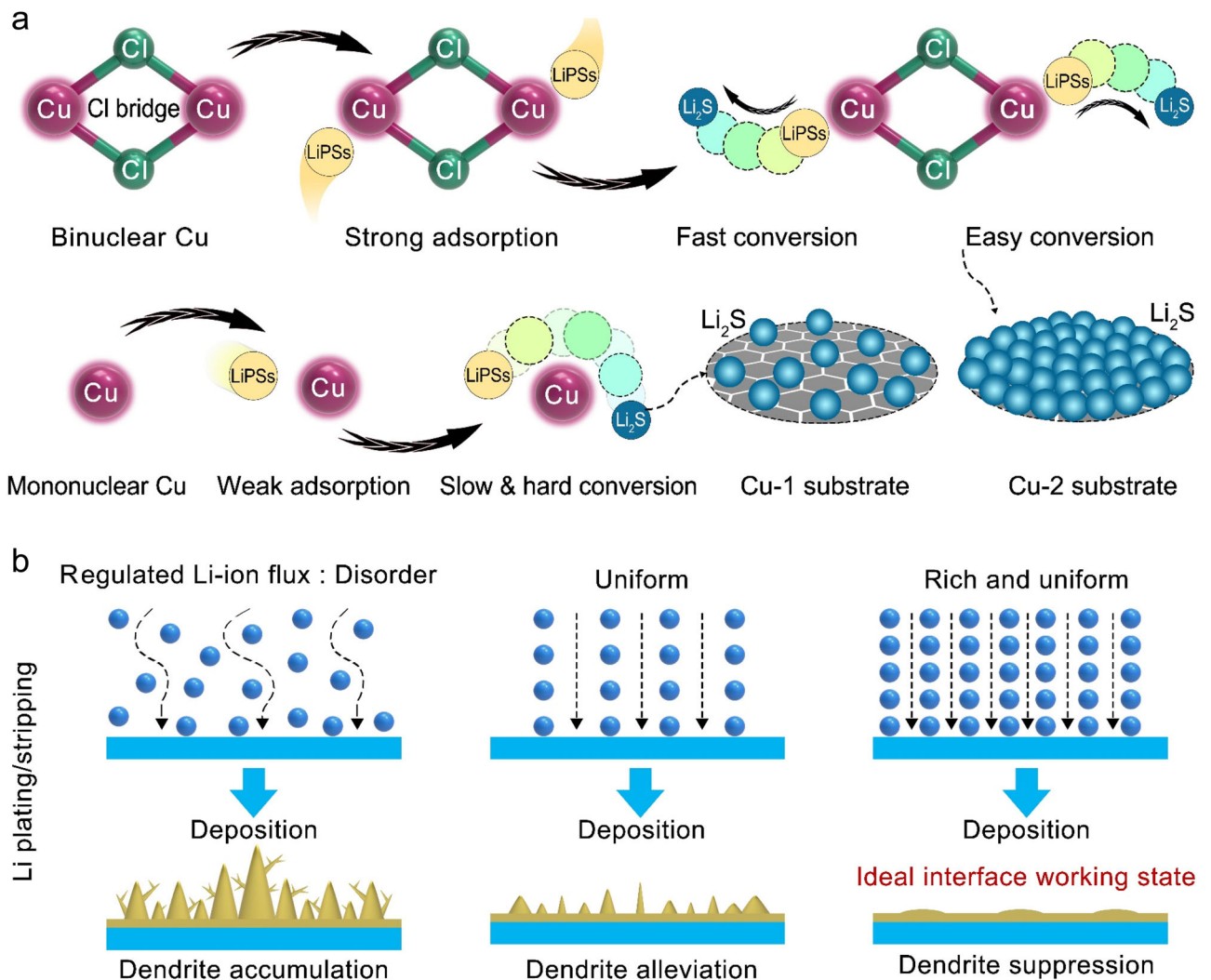

**Fig. 2 | Schematic illustrating the catalytic effect towards LSBs. a** LiPS conversions catalyzed by Cu-1 and Cu-2. **b** Lithium evolutions manipulated by Cu-1 and Cu-2.

density distribution of the whole ligand, hence leading to the shift of all the peaks. The atomic structures of 1-Cuphen and 2-Cuphen by single crystal X-ray diffraction (XRD) tests are shown in Fig. 3c, e (CCDC: 2217502 and 2047772). The detailed bond lengths and angels are displayed in Tables S1 and S2. In Fig. 3c, two Cu atoms are bridged by two Cl atoms in 2-Cuphen molecule to form the symmetrical binuclear Cu sites. The spherical aberration corrected transmission electron microscope (TEM) images show the crumpled graphene morphologies of Cu-1 and Cu-2 (Figs. S3, S4). High-angle annular dark-field scanning TEM (HAADF-STEM) images in Fig. 3d, f reveal that the Cu single atoms are uniformly distributed on the substrates for both Cu-2 and Cu-1. Furthermore, Cu atoms in Cu-2 are in pairs (marked with red circles), proving the formation of binuclear Cu sites. The distances between the proximal two Cu atoms in the HAADF-STEM image of Cu-2 are around 0.35 nm, in accordance with the results of single crystal XRD tests (Fig. S5a, b). For the Cu-1 sample, the calculated distances between the neighboring two Cu atoms are obviously fluctuant (Fig. S5c, d). The above results demonstrate that the Cl-bridge pairs maintain the structural stability of proximal two Cu atoms in Cu-2 sample. Inductively coupled plasma optical emission spectrometer (ICP-OES) tests were conducted to experimentally probe the content of Cu centers in Cu-1 and Cu-2. According to the measurement results, the Cu contents in Cu-1 and Cu-2 are 1.16 and 1.18 wt%, respectively, substantiating the similar contents of Cu centers in the two catalysts.

The phase structures of 1-Cuphen and 2-Cuphen by powder XRD measurements are in line with the simulated data stemming from the single crystal XRD defined results (Fig. 3g, h). In comparison, there is no characteristic peaks of 1-Cuphen and 2-Cuphen existing in the XRD patterns of Cu-1 and Cu-2, respectively, indicating the molecule dispersion of 1-Cuphen and 2-Cuphen on graphene (Fig. S6). The scanning TEM (STEM) images of samples are displayed in Fig. S7. And the elemental maps of Cu, N, Cl, and C demonstrate that the 1-Cuphen and 2-Cuphen molecules are homogeneously distributed onto the graphene substrates (Fig. 3i, j). X-ray photoelectron spectroscopy (XPS) was utilized to explore the chemical compositions of the two electrocatalysts (Figs. 3k and S8–10). It can be clearly witnessed the signals of bridged and nonbridged Cl atoms in Cu-2 with the corresponding binding energy as 198.2 and 197.8 eV for Cl $2p_{3/2}$, respectively, which is quite different with those for Cu-1 (Fig. S8). The Cu $2p$ spectrum of Cu-2 displays a lower binding energy of 934.8 eV than Cu-1 (934.9 eV), indicative of their different coordination environments. Finally, the $N_2$ adsorption-desorption isotherms were applied to obtain the Brunauer–Emmett–Teller (BET) surface areas of samples. And the BET surface area values of phen loaded graphene (rphenGO), Cu-1, and Cu-2 are 91.0, 103.4, and 93.8 $m^2 g^{-1}$, respectively (Fig. S11).

LiPS adsorption capability is the prerequisite for the shuttle effect alleviation. In this case, visualized $Li_2S_4$ adsorption experiments were first executed to probe the adsorption properties of Cu-2 and Cu-1. As observed in Fig. S12, the $Li_2S_4$ solutions in vials with

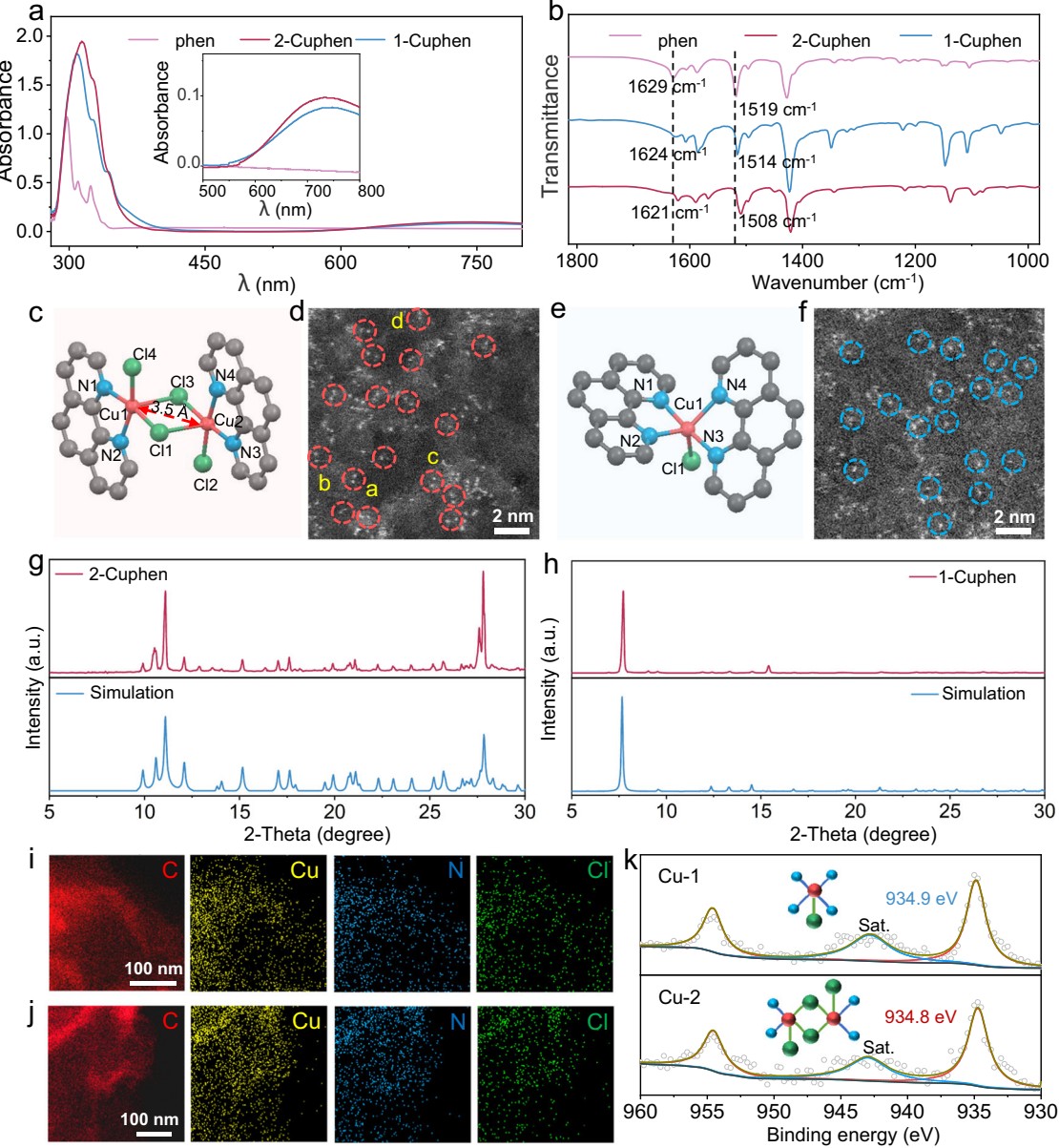

**Fig. 3 | Morphological and structure characterizations of electrocatalysts.** **a** UV–vis and **b** FTIR spectra of phen, 1-Cuphen and 2-Cuphen. **c, e** Molecular structure of 2-Cuphen and 1-Cuphen. **d, f** Atomic-resolution HAADF-STEM images of Cu-2 and Cu-1. **g, h** Experimental and simulated XRD patterns of 2-Cuphen and 1-Cuphen. **i, j** TEM elemental mapping images of Cu-2 and Cu-1. **k** Cu 2p XPS spectra of Cu-2 and Cu-1.

rphenGO and Cu-1 as the adsorbers turn light yellow after 10 h. As a contrast, Cu-2 almost decolor the $Li_2S_4$ solution after 10 h, indicative of its stronger LiPS adsorption capability. According to the UV–vis spectra, the absorption signal of $Li_2S_4$ at ~420 nm disappears due to the superior Cu-2 capture capability in the sharp contrast with the scenarios of rphenGO and Cu-1 samples[47]. To further probe the affinity of $Li_2S_4$ by Cu-1 and Cu-2, XPS was employed to inspect the chemical compositions of samples after the visualized adsorption measurements. As displayed in Fig. S13, new peaks of Cu $2p_{3/2}$ are detected at lower binding energies for Cu-2 (932.4 eV) and Cu-1 (932.5 eV), indicating that Cu atoms can serve as the active centers for anchoring LiPSs. More importantly, the peaks of Cu $2p_{3/2}$ shift by 2.4 eV and the new peak percentage are around 60% for both Cu-1 and Cu-2. Note that only one new peak appears in the Cu $2p_{3/2}$ spectrum of Cu-2, manifesting that the dual Cu atoms in Cu-2 act as double-adsorption sites, rather than a one center for anchoring $Li_2S_4$ species. The spectrum of S $2p$ shows peaks centered at 162.2 and

162.4 eV which can be assigned to $2p_{3/2}$ of Cu–S bond, further confirming the effective anchoring effect of Cu-2 and Cu-1 with $Li_2S_4$.

Operando Raman tests were employed to track the evolution behavior of sulfur species in a real-time manner. The cathode was obtained by the disconnected coverage of slurry on a piece of Al mesh with a pore size of 0.5 mm × 0.1 mm. After drying, the cathode was cut into small disks with a diameter of 13.0 mm for the operando Raman cell fabrication. Note that the disconnected material coverage simultaneously provides the cathode and catholyte regions for the operando Raman signal collection. Figure 4a displays the operando Raman spectra of cathode and catholyte regions on separator throughout the complete discharge/charge procedure at 0.2 C. With respect to the cathode region, the peaks at ~219 and ~473 $cm^{-1}$ are assigned to octatomic $S_8$, while the sharp peak at ~153 $cm^{-1}$ is ascribed to $S_8^{-2}$ [48,49]. These Raman signals of $S_8$ and $S_8^{-2}$ gradually decrease in the discharge procedure and then increase in the charge process, reflecting the dynamic changes of sulfur species on the surface of

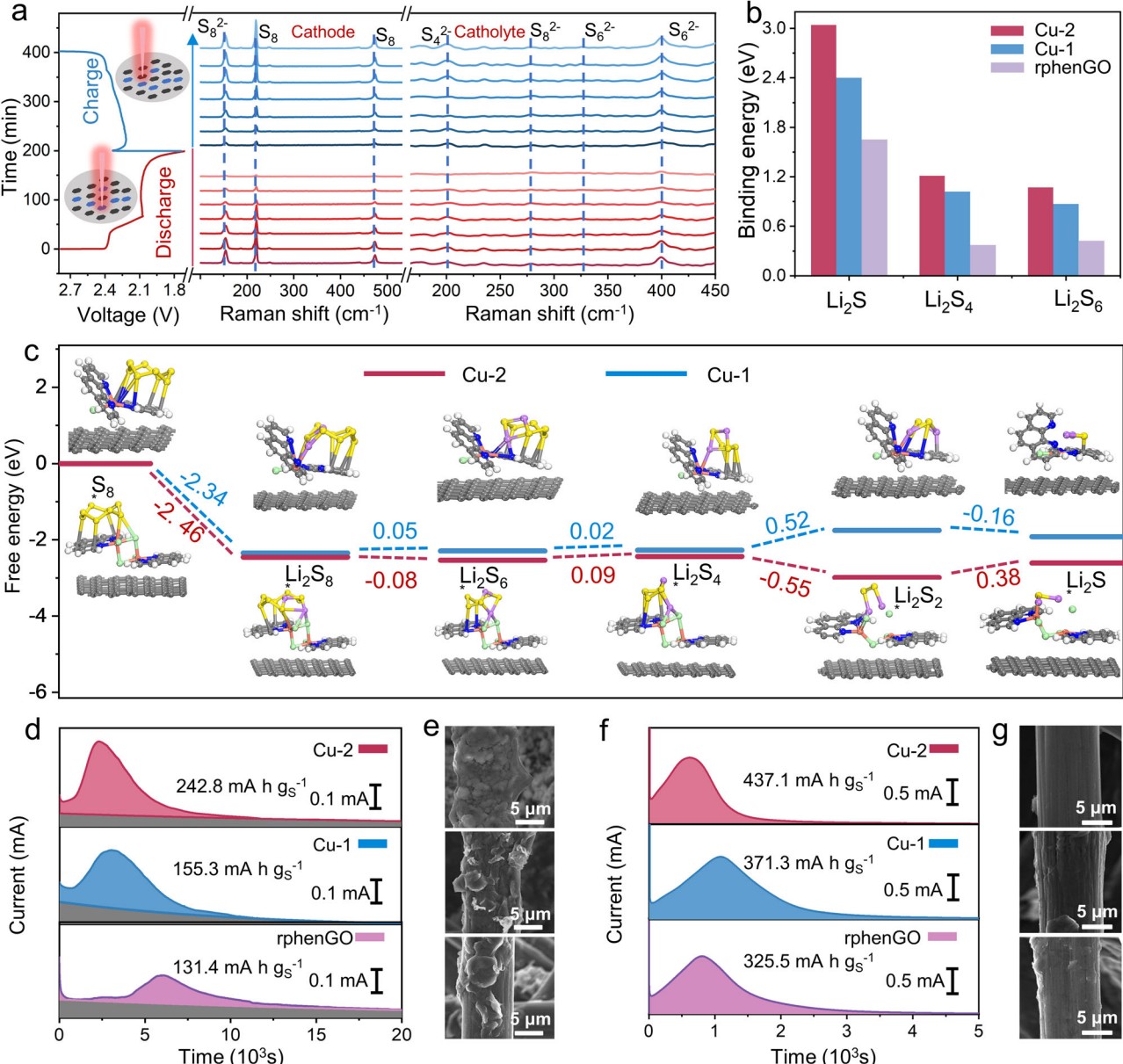

**Fig. 4 | Electrochemical tests, spectrum characterizations and calculation simulations of sulfur species evolution on various electrocatalysts. a** Operando Raman spectra of S/Cu-2 cathode and catholyte on separator during the discharge/charge process at 0.2 C. **b** Binding energies of sulfur species on various substrates. **c** Gibbs free energy profiles for the sulfide conversion reactions on Cu-2 and Cu-1. **d** Potentiostatic discharge curves of Li$_2$S$_8$/tetraglyme solution on Cu-2, Cu-1 and rphenGO substrates, respectively. **e** SEM images of deposited Li$_2$S. **f** The dissociation curves of Li$_2$S on Cu-2, Cu-1 and rphenGO substrates. **g** SEM images of electrochemically dissociated Li$_2$S.

Cu-2. The Raman signals of soluble LiPSs were also tracked in the catholyte region. In detail, the signals of S$_4^{-2}$ (~202 cm$^{-1}$), S$_6^{-2}$ (~327 and ~399 cm$^{-1}$) and S$_8^{-2}$ (~279 cm$^{-1}$) presents the same variation trend due to the management of Cu-2 during the discharge/charge process. Operando Raman spectra of catholyte regions with respects to Cu-1 and rphenGO were also shown in Fig. S14. The catholyte region of S/Cu-2 shows the weaker soluble LiPS Raman signals along with the discharge proceeding in contrast to the other two samples. Particularly, the Raman signals of LiPSs in catholyte region of S/Cu-2 almost disappear at the end of discharge. These results further confirm the stronger LiPS anchoring ability of Cu-2[50,51]. The operando Raman test results reveal the effective manipulation of the dynamic evolution behaviors of LiPSs by Cu-2, implying the favorable functions of Cu-2 on the shuttle effect mitigation and redox reaction catalysis. Theoretical simulations based on density functional theory

were performed to verify the interaction between complexes and sulfur species. In Fig. 4b, the binding energies between Cu-2 and Li$_2$S$_6$, Li$_2$S$_4$ and Li$_2$S are 1.07, 1.21, and 3.04 eV, respectively, the values of which are much higher than those of Cu-1 (0.87, 1.02, and 2.40 eV) and rphenGO (0.42, 0.37 and 1.65 eV), substantiating the strong LiPS capture capability of Cu-2. The optimal adsorption configurations and other binding energy values for the three samples are displayed in Figs. S15–17. The evolution pathways of sulfur species on the electrocatalyst substrates and the related Gibbs free energy values are exhibited in Figs. 4c and S18. Notably, the negative Gibbs free energy change (Δ$G$) for the conversion of solid S$_8$ to liquid Li$_2$S$_8$ represents the thermodynamically spontaneous exothermic step. For the Cu-2, Cu-1, and rphenGO, the Δ$G$ values for the rate-determining step are 0.38, 0.52, and 0.75 eV, respectively, indicating the more easily-occurred sulfur species reduction on Cu-2[52].

Chronoamperometry measurements were first conducted to evaluate the kinetics of Li₂S precipitation and dissociation reactions on the surface of catalyst substrates. As depicted in Fig. 4d, the capacity of Li₂S precipitation on Cu-2 at 2.05 V is 242.8 mA h g$_s^{-1}$, the value of which is much higher in contrast to that on Cu-1 (155.3 mA h g$_s^{-1}$) and rphenGO (131.4 mA h g$_s^{-1}$), implying the remarkable electrocatalytic activity of Cu-2 for the Li₂S nucleation reaction. The cells were disassembled after the nucleation procedure to examine the morphology and distribution of Li₂S on substrates. It can be witnessed in Fig. 4e that the Cu-2 substrate fulfills the most complete coverage and homogeneous deposition of Li₂S among the three samples, further demonstrating the effectively propelled Li₂S nucleation and growth kinetics on Cu-2. In addition, the decomposition profiles of Li₂S on substrates were recorded by a potentiostatic charging the cells at 2.35 V. As shown in Fig. 4f, the capacities of Li₂S dissociation on Cu-2,

Cu-1, and rphenGO are 437.1, 371.3 and 325.5 mA h g$_s^{-1}$, respectively. Whilst, scanning electron microscopy (SEM) images of electrode after the dissociation reaction exhibit the cleaner surface than other two samples (Fig. 4g). Note that the higher activity for the Li–S redox reactions is harvested by Cu-2 than other samples owing to the identified homonuclear dual Cu centers according to the above potentiostatic test results. The proximal binuclear Cu centers in Cu-2 as well as their optimized coordination environment provide more active interfaces for realizing the appropriate electrokinetic control of the Li₂S nucleation and decomposition reactions, manifesting the kinetically promoted discharge and charge processes of the practically operated LSBs.

Cyclic voltammetry (CV) tests were further performed to probe the catalytic conversion process of LiPSs based on various catalysts. Figure 5a manifests the CV profiles of symmetrical cells within the

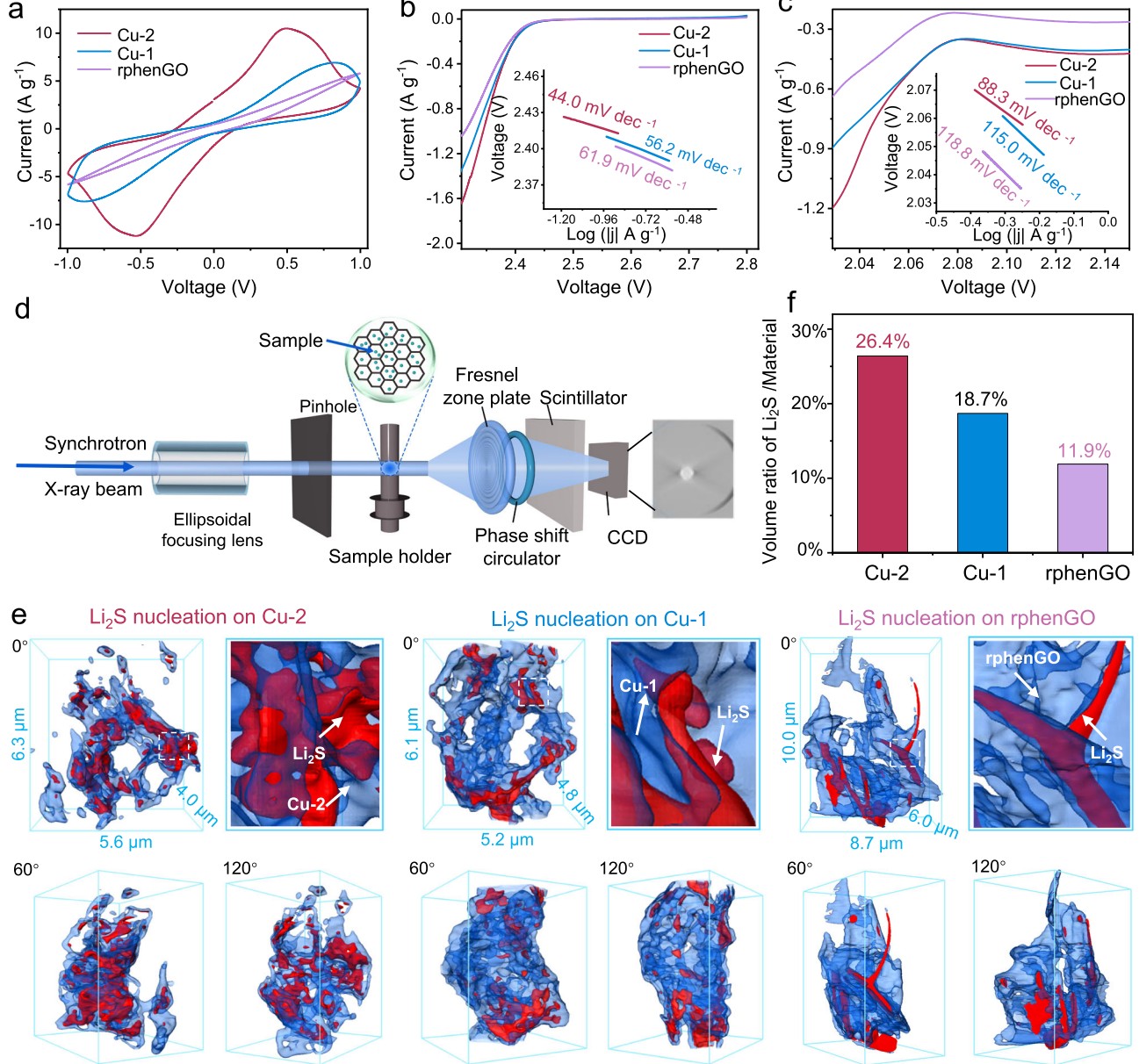

**Fig. 5 | The electrocatalytic activity analysis of electrocatalysts for sulfur conversion reactions. a** CV profiles of Cu-2, Cu-1 and rphenGO based symmetric cells. **b, c** LSV curves with corresponding Tafel plots showing in the insets. **d** Schematic diagram of X-ray 3D nano-CT facility for sample observation. **e** X-ray 3D nano-CT images of Li₂S precipitations on Cu-2, Cu-1 and rphenGO substrates. **f** Statistic volume ratios of Li₂S deposits on various substrates based on X-ray 3D nano-CT analysis.

voltage window of −1 to 1 V at a scan rate of 50 mV s$^{-1}$. It's apparent that CP/Cu-2 shows higher redox peak current than CP/Cu-1 and CP/rphenGO, suggesting the enhanced catalytic activity of Cu-2 toward LiPS redox reactions[38]. With a lowered sweeping rate of 0.5 mV s$^{-1}$, two pairs of reversible redox peaks of the CV curves in Fig. S19 are labeled as a, b, c, and d. Peak a and d stand for the reduction conversion of soluble LiPSs to Li$_2$S. Peak b and c respectively represent the reversible procedure of peak d and a[14]. It can be clearly observed that the CP/Cu-2 exhibits the smaller polarization compared to CP/Cu-1 for the whole sulfur redox reaction. Linear sweep voltammetry (LSV) measurements were also carried out with S/Cu-2, S/Cu-1 or S/rphenGO as the cathode and lithium foil as the anode to gather the Tafel slope values. As shown in Fig. S20, there are two clear reduction peaks at 2.2–2.4 V (peak I) and 1.9–2.1 V (peak II) in the LSV curves, corresponding to the reduction of S$_8$ to soluble Li$_2$S$_n$ ($4 \leq n \leq 8$) and the subsequent conversion to insoluble Li$_2$S$_2$/Li$_2$S, respectively. For S/Cu-2, the positive shift of the peak I and II in the LSV curve reveals the facilitated sulfur reduction process in contrast to the other two cathodes. Moreover, the fitting Tafel slope values of S/Cu-2 for peak I and II are 44.0 and 88.3 mV dec$^{-1}$, respectively, the values of which are lower than those of S/Cu-1 (56.2 and 115.0 mV dec$^{-1}$) and rphenGO (61.9 and 118.8 mV dec$^{-1}$), implying more favorable conversions on Cu-2 (Fig. 5b, c)[53]. The activation energy ($E_a$) for each conversion reaction step of sulfur intermediates to further reflect the catalytic activity of Cu-2. Electrochemical impedance spectra (EIS) tests under different temperatures were performed at various voltages where critical sulfur reactions occur (Fig. S21). Particularly, Nyquist plots at 2.4 and 2.1 V respectively represent the conversion step of soluble LiPSs and Li$_2$S$_4$ to Li$_2$S$_2$/Li$_2$S. After fitting the circuits, Arrhenius equation was applied to calculate the values of $E_a$ (Fig. S22). As displayed in Fig. S22d, the S/Cu-2 cathode obtains the lowest the values of $E_a$ for each voltage among the three samples, further corroborating the high catalytic activity of Cu-2 for the whole sulfur conversion reaction.

Apart from the information from the surface of cathodes, the inner of cathodes more truly reflect the situations of Li$_2$S nucleation reactions. Synchrotron radiation X-ray 3D nano-CT as a powerful imaging facility was employed to accurately evaluate the information of Li$_2$S deposit in the inner of cathodes[54]. The catalyst/Li$_2$S samples were acquired from the disassembled cells after the Li$_2$S nucleation measurements. The 2D tomograms of samples based on various rotation angles were reconstructed to produce the final 3D visualized images (Fig. 5d). According to the imaging results in Figs. 5e and S23, the Li$_2$S precipitations on Cu-2 exhibits more homogeneous spatial distribution than that on Cu-1 and rphenGO. The quantitative results of Li$_2$S deposit on catalysts are displayed in Fig. 5f. As seen in Fig. 5f, the volume ratio of Li$_2$S and Cu-2 is 26.4%, which is much larger than that on Cu-1 (18.7%) and rphenGO (11.9%), in accordance with the results of chronoamperometry measurements at 2.05 V. The synchrotron radiation 3D X-ray nano-CT results disclose that Cu-2 produces superior electrocatalytic effect on the whole cathode involving the surface and inner, which is beneficial to attaining the favorable electrochemical performance. These results confirm the proximal distance of Cu active sites and well-designed coordination structure of Cu-2 which enable the superior efficiency of Li$_2$S conversion reaction on Cu-2.

The lithium stripping and plating behaviors were explored by fabricating the Li//Li symmetric cells with catalyst coated on the commercial Celgard separator (Fig. S24). As depicted in Figs. 6a and S25, at a current density of 1.0 mA cm$^{-2}$ and a capacity of 1.0 mA h cm$^{-2}$, the Li//Li symmetric cell inserted with bare commercial PP separator presents dramatic voltage fluctuations due to the uneven Li deposition and severe dendrite growth. As a contrast, Cu-2 electrocatalyst endows the cell with stable lithium plating/stripping overpotentials and prolonged cycling lifespan over 1100 h, surpassing the Cu-1 coated and pristine Celgard separator. Moreover, Li//Li

symmetric cell with Cu-2 remains a stable voltage hysteresis of 11 mV, which is slightly lower than that of Cu-1 (inset of Fig. 6a). The EIS of Li//Li symmetric cells were collected after cycling at 1.0 mA cm$^{-2}$. The charge transfer resistance of cycled cell with Cu-2 is less than ~5.0 ohm, in sharp contrast to the ~70.0 ohm of the cell inserted with commercial PP separator (Fig. 6b). Figure S26 illustrates the rate performance of the Li//Li symmetric cells with various separators under the current density varying from 1 to 3 mA cm$^{-2}$ with a fixed capacity of 1 mA h cm$^{-2}$. In comparison with other separators, Cu-2 coated separator enables the Li//Li symmetric cell to exhibit stable operation at different current densities. Although Cu-1 coated separator boosts the rate cycling performance of Li//Li symmetric cell than the commercial Celgard separator, short circuit phenomenon still appears under a high current density of 3 mA cm$^{-2}$. The superior cycling stability and rate performance are harvested by the Li//Li symmetric cell inserted with Cu-2 coated separator, verifying that the binuclear Cu sites are conduced to forming stable solid electrolyte interface (SEI) film and homogenizing the lithium metal deposition.

Coulombic efficiency (CE) is an important parameter to evaluate the reversibility of lithium plating/striping. Along this line, the CE values were collected by assembling Cu//Li cells with Cu severing as the working electrode and Li metal as the counter electrode. As shown in Fig. 6c, the asymmetric cell inserted with Cu-2 coated separator still obtains a CE value of 98.9% after 150 cycles, while the CE value for Cu-1-enabled cell is only 97.6% during 113 cycles. Pristine commercial separator only maintains stable operation for 70 cycles with the CE value as 96.7%. The lower CE may be attributed to the uneven and loose SEI film which is easily penetrated by the dendrites. Figure S27 exhibits the lithium plating curves of Cu//Li cells at the first cycle where the nucleation overpotentials, pertaining to the gap between the voltage dip and the voltage plateau, are quite different. Apparently, Cu-2 endows the cell with a nucleation overpotential of 31.7 mV, the value of which is lower than Cu-1 coated (34.1 mV) and pristine separator (87.1 mV) at 1 mA cm$^{-2}$. To uncover the underlying mechanism for the superior lithium protection effect of Cu-2, the surface chemical compositions of lithium foil after cycling were investigated by XPS (Figs. 6d and S28). The peaks located at 56.3, 55.6, and 53.7 eV in Li 1$s$ spectra are ascribed to LiF, Li$_2$CO$_3$ and Li$_2$O, respectively. In addition, the peak at 54.5 eV is assigned to Li$_2$O$_2$ or LiOH which are unstable ingredients for SEI film[55]. Figs. 6e and S29 further demonstrate the total contents of LiF, Li$_2$O$_2$, Li$_2$O and LiOH analyzed by the Li 1$s$ XPS fitting data. The higher LiF ratio (30.3%) manifests the more stable SEI film in the Cu-2-enabled cell as compared to that of Cu-1 coated (14.0%) and pristine PP separator (4.1%).

The structures of lithium anodes after stripping and plating procedures were further detected by SANS and SEM. The cavity stemming from the nonuniform lithium deposition at anode has a trend to display strong scattering signals. The scattering intensity I(Q) to scatter vector Q plots are fitting in SASfit software with model independent analysis involving Guinier and Porod approximations. And the gyration radius ($R_g$) can be calculated from the scattering curves. The value of $R_g$ is in the order of Cu-2 < Cu-1 < PP, which suggests that the Cu-2 separator enables more dense lithium deposition than Cu-2 and PP. The results are in accordance with the depositing morphologies detected by SEM (Fig. 6f–h, insets). As expected, the surface of lithium foil is uneven, angular and loose for the cell inserted with pristine PP separator. On the contrary, the morphology of lithium foil with regard to the cell inserted with Cu-2 coated separator is more smooth and dense, further corroborating the effective lithium dendrite inhibition by Cu-2. Furthermore, the morphology of anode and the concentration of electrolyte during the deposition process were calculated based on finite element analysis (Fig. 6i, j). The results show Cu-2 endows the anode with a relatively dense deposition without obvious dendrite formation, indicating its better dendrite suppression ability. By contrast, the concentration gradient of lithium ions for initial

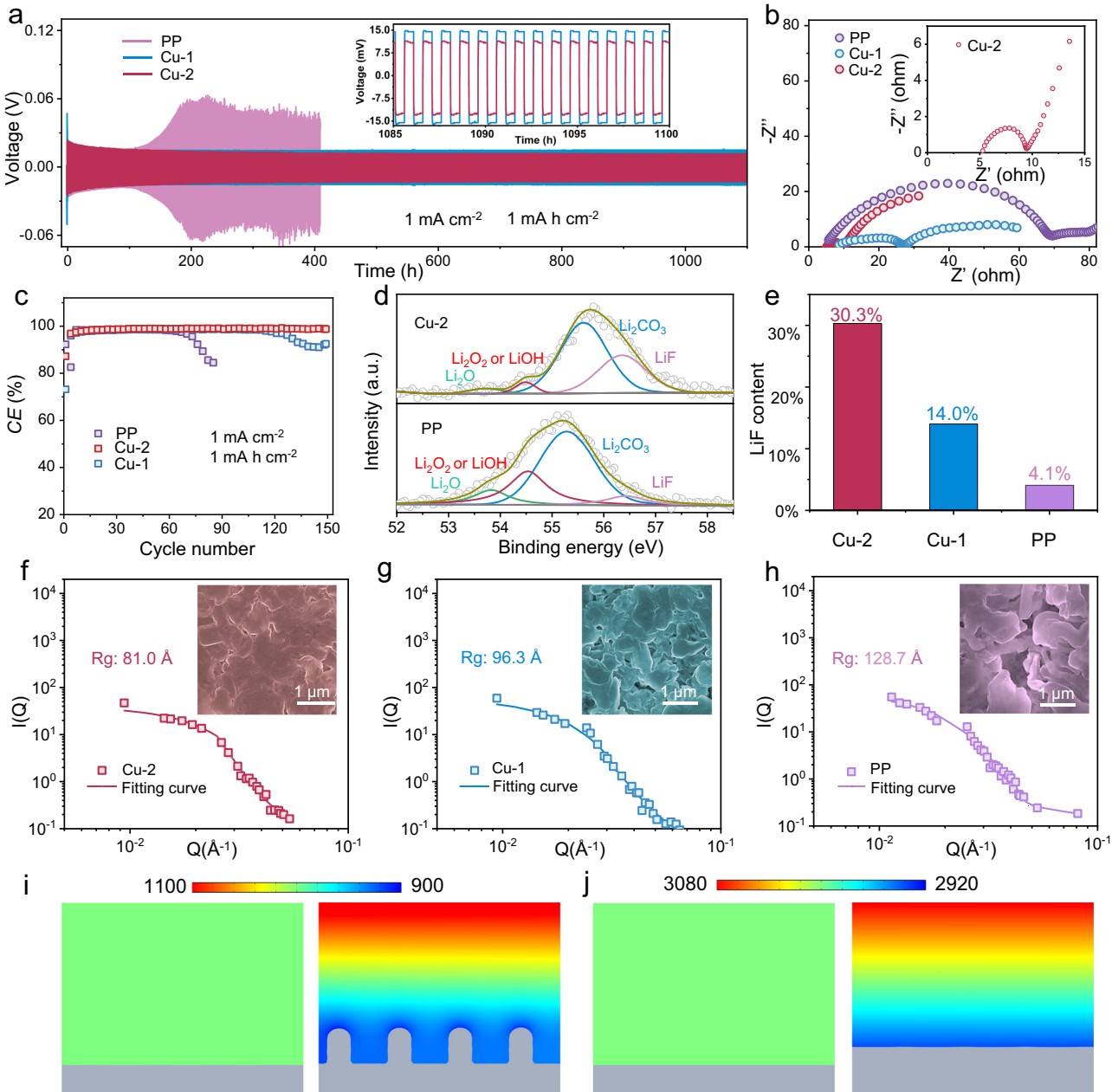

**Fig. 6 | Effect of electrocatalysts on lithium anodes. a** Cycling performance of Li// Li symmetric cell with the Cu-1,Cu-2 coated separators and pristine separator at 1.0 mA cm⁻². **b** EIS curves for Li//Li symmetric battery after cycling. **c** CE of Cu//Li cells with different separators at 1 mA cm⁻². **d** Li 1s XPS fitting data with etching for cycled sample. **e** LiF content statistics from Li 1s XPS fitting data. **f–h** SANS matrix of scattering intensity signals and fitting curves from Li anodes in Li//Li symmetric cells with Cu-2, Cu-1, and PP separators after cycling. COMSOL results of Li plating with PP (**i**) and Cu-2 (**j**) coated separators.

sample and Cu-1 sample (Fig. S30) are smaller, and the dendrites are much serious. These results substantiate that the Cl bridge bond-enabled Cu-2 acts as favorable regulator in lithium stripping and plating procedures.

The coin-type LSBs with S/Cu-2, S/Cu-1, S/rphenGO, and S/rGO cathodes were respectively assembled to test their electrochemical performance. The rate capacities were first measured and the results are shown in Fig. 7a. At 0.2, 0.5, 1.0, and 2.0 C, the S/Cu-2 cathode delivers reversible capacities of 1121.1, 1028.8, 936.4, 808.6 mAh g⁻¹, respectively. When the discharge current is switched back to 0.2 C, a high reversible discharge capacity of 1100.1 mAh g⁻¹ is recovered, indicative of the rate capability of S/Cu-2 cathode. For the S/Cu-1 cathode, the discharge capacities are 1100.2, 959.6, 853.6, and 755.8 mAh g⁻¹ in the same rate order and the recovered capacity at

0.2 C is 981.7 mAh g⁻¹. The S/rphenGO delivers the capacities of 1034.3, 737.9, 554.4, and 400.3 mAh g⁻¹ and the S/rGO presents the lowest rate capacities (819.3, 574.1, 334.9, and 200.0 mAh g⁻¹) at 0.2, 0.5, 1.0 and 2.0 C. Moreover, the reaction polarization is assessed by the overpotential calculated by the voltage gap from the charge/discharge profiles (Figs. 7b and S31). The lowest voltage gaps at various rates of S/Cu-2 among all the samples further manifest the effectively regulated polarization by Cu-2. The cycling performance of cathodes at 0.2 C was evaluated. In Figs. 7c–e and S32, the S/Cu-2 cathode delivers a higher initial capacity of 1140.6 mAh g⁻¹ and a more remarkable capacity retention of 91.3% over 100 cycles in contrast with those of S/Cu-1 (1062.0 mAh g⁻¹ and 80.0%), S/rphenGO (1049.3 mAh g⁻¹ and 71.1%) and S/rGO (802.7 mAh g⁻¹ and 74.7%), validating its favorable cycling performance. The cathodes after 100

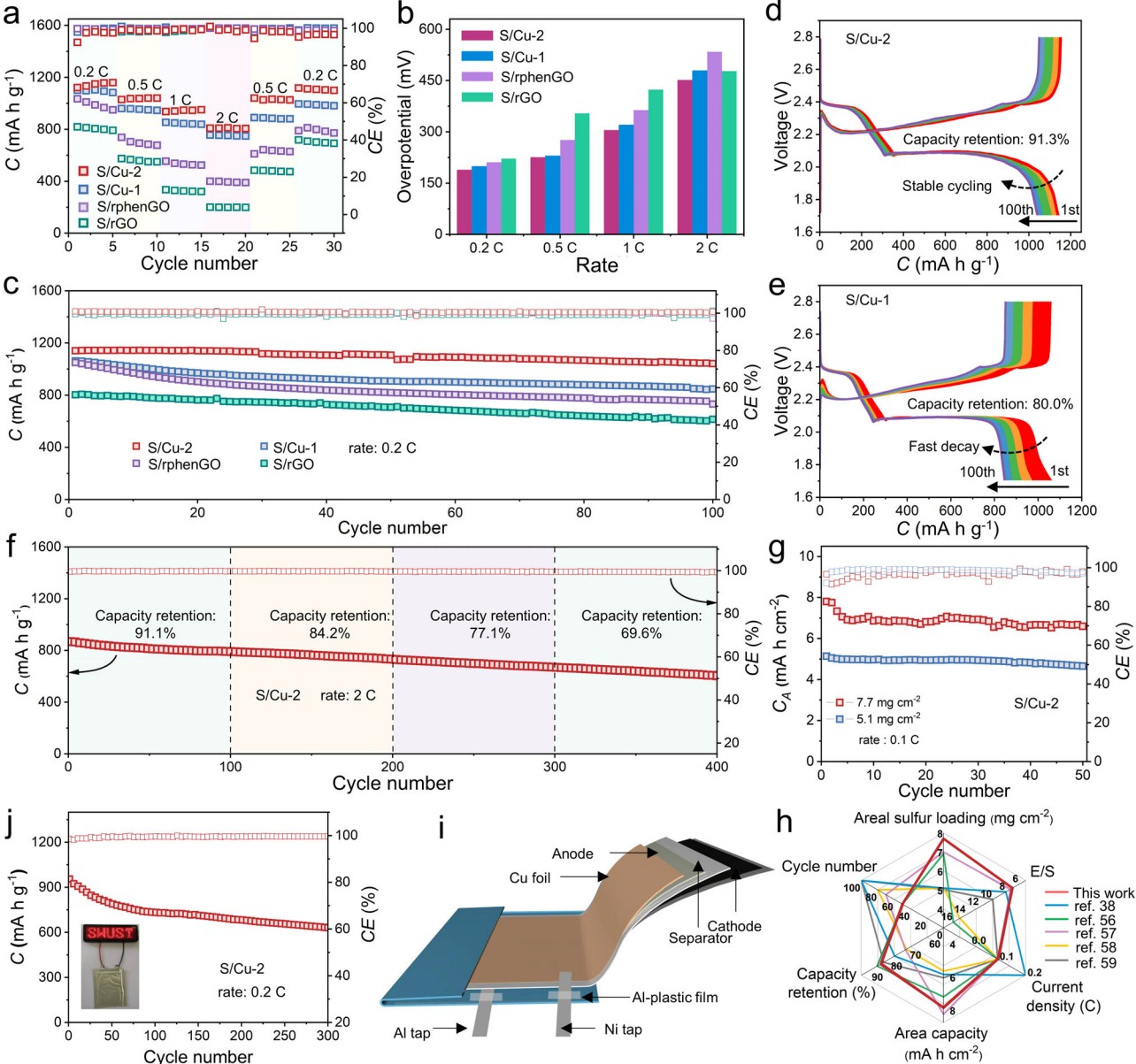

**Fig. 7 | Electrochemical performance of LSBs with various cathodes. a** Rate capabilities of S/Cu-1, S/Cu-2, S/rphenGO and S/rGO cathodes. **b** Voltage gaps of S/Cu-1, S/Cu-2, S/rphenGO and S/rGO cathodes at various rates. **c** Cycling stability of the cathodes at 0.2 C. **d**, **e** Galvanostatic discharge-charge curves of different cycles at 0.2 C for S/Cu-2 and S/Cu-1. **f** Cycling performance of S/Cu-2 cathode at 2 C. **g** Cycling performance of the S/Cu-2 cathode with the high-loading sulfur. **h** High-sulfur-loading LSB performance comparisons between this work and other reports. **i** Schematic diagram of S/Cu-2-enabled pouch cell. **j** Long-term cycling performance of the S/Cu-2 cathode at 0.2 C for pouch cell.

cycles at 0.2 C were inspected by SEM. It can be seen that both the S/Cu-2 and S/Cu-1 cathode maintain better structural integrity and robustness while obvious cracks exist in the S/rphenGO and S/rGO cathode (Fig. S33). Furthermore, the S/Cu-2 cathode holds the capacity retentions of 91.1%, 84.2%, 77.1%, and 69.6%, respectively, when working at 100, 200, 300, and 400 cycles, corroborating the remarkable optimization impact of binuclear Cu centers in Cu-2 on the evolution behaviors of sulfur and lithium species (Fig. 7f).

The Cu-2 was further applied in the practical scenarios of high sulfur loading and flexible soft-packaged pouch cells. As displayed in Figs. 7g and S31, with a sulfur loading of 5.1 and 7.7 mg cm$^{-2}$, the S/Cu-2 cathode can respectively achieve the superior initial areal capacities of 5.1 and 7.8 mA h cm$^{-2}$, and retentions of 90.6% and 84.6% at 0.1 C over 50 cycles, verifying the remarkable catalytic effect of Cu-2. The S/Cu-2 cathode with the sulfur loading of 7.7 mg cm$^{-2}$ exhibits more favorable electrochemical performance in contrast to the previously reported high-loaded cathodes, as seen in Table S3 and Fig. 7h[38,56–59]. The high sulfur loading and low electrolyte usage are of utmost importance to the practically implementation of LSBs. Therefore, the cycling lifespan of the high-loaded cathodes need be further optimized towards commercially viable LSBs throughout unremitting endeavor from the comprehensive design of catalysts, electrolytes, binders, etc. Further, the soft-packed pouch cell with the size of 3 cm × 3 cm was fabricated (Fig. 7i). As witnessed in Figs. 7j and S35, the as-achieved cell can not only power a light-emitting diode device, but also realizes the stable cycling over 300 cycles (0.11% capacity decay per cycle). Such favorable electrochemical performance under pragmatic working conditions showcases the great application potential of binuclear copper complex on achieving advanced LSBs with high energy density and flexibility.

## Discussion

In summary, the homonuclear dual copper atom complex with remarkable activity is developed by constructing Cl bridge bonds toward advanced LSBs. The well-designed catalyst can make full use of the binuclear copper centers with a proximal distance of 3.5 Å to increase the active interface concentration. Such proximity effect of Cu-2 guides the $Li_2S$ nucleation and decomposition reactions, and manipulate the lithium stripping and plating behaviors, boosting the durable performance and safety of LSBs under really-operated conditions. Therefore, at a high sulfur mass loading of 7.7 mg cm$^{-2}$, the LSB still obtains a remarkable areal capacity of 7.8 mA h cm$^{-2}$ as well as high retention of 84.6%. Impressively, the as-designed soft-packaged pouch cell can maintain the long-term stability for 300 cycles. Therefore, this dual-atom electrocatalyst harnessing Cl bridge linker offers a feasible strategy to concurrently tackle the issues of shuttle effect and dendrite growth in the pursuit of efficient and stable LSBs.

## Methods

### Preparation of Cu-2

1,10-phenanthroline (1 g, 5 mmol) and copper(II) chloride dihydrate (0.85 g, 5 mmol) were added in a 25 mL Teflon-lined autoclave (Anhui CHEM$^N$ Co. Ltd.), and 10 mL ultra-pure water mixed with 1 mL ethanol used solvent. The mixture was maintained under 150 °C for 24 h. After cooled down to room temperature, green crystal was collected by filtration and washing with icy isopropyl alcohol and water mixture which was denoted as 2-Cuphen. 2-Cuphen was dispersed in 20 mL ethanol to form dispersion A, which was dropwise added to reduced graphene oxidation (rGO) dispersion B. Then the mixture was stirred at room temperature overnight before filtration and washing with ultra-pure water. The Cu-2 was obtained after a freeze-drying procedure.

### Preparation of Cu-1

To prepare Cu-1, the mole ratio of 1,10-phenanthroline and copper (II) chloride was adjusted to 1:2. The solvent was *n*-butanol. Apart from that, the synthesis of Cu-1 was the same as Cu-2. The Cu contents in Cu-1 and Cu-2 are 1.16 and 1.18 wt%, respectively, detected by ICP–OES.

### Material characterizations

The morphologies of materials were collected on Zeiss sigma500 SEM and JEOL-JEM 2100 F transmission electron microscopy (TEM). High-angle annular dark-field scanning transmission electron microscopy (HAADF-STEM) images and elemental maps of materials were recorded on a JEOL JEM 200F TEM/STEM equipped with a spherical aberration corrector. The single crystal structure data were collected on a Bruker SMART APEX-II CCD detector system. Power XRD patterns of samples were achieved on a PANalytical X'Pert Pro diffractometer. ICP-OES (Agilent 725) was used to detect the Cu contents. Nitrogen adsorption-desorption isotherms were recorded at 77 K in $N_2$ atmosphere on a Micromeritics Autosorb-IQ Accelerated Surface Area and Porosimetry System. A Thermo Scientific K-Alpha+ X-ray photoelectron spectroscope was applied to probe the surface chemical compositions of samples. UV–vis spectra were collected on an UV–vis spectrometer (Shimadzu, UV1900). In situ Raman spectra of LSBs were collected on a Horiba LabRAM HR Evolution confocal Raman instrument. A LSBs-Raman cell (Beijing Scistar Tech. Co. Ltd.) was used in the operando Raman measurement.

### $Li_2S$ nucleation/ decomposition tests

0.2 mol L$^{-1}$ $Li_2S_8$ solution was first attained by mixing sulfur and $Li_2S$ at a molar ratio of 7:1 in tetraglyme solvent. CP was punched into small disks with a diameter of 13 mm. Cu-1, Cu-2 and rphenGO powder were dispersed into ethanol and then loaded on CP. Cells were fabricated by employing CP-Cu-1 or CP-Cu-2 as the cathode and lithium foil as the anode, 20 μL $Li_2S_8$ solution as the catholyte and 20 μL LiTFSI (1.0 mol L$^{-1}$) solution without $Li_2S_8$ as the anolyte. The assembled cells were first galvanostatically discharged to 2.06 V at a current of 0.112 mA and then discharged potentiostatically at 2.05 V until the current was below 10$^{-5}$ A. The nucleation capacity of $Li_2S$ was calculated by the integral area of the plotted curve based on Faraday's Law. For the $Li_2S$ decomposition tests, the cells were fabricated by following the same assemble process as the nucleation test. To ensure the complete transformation of LiPSs into $Li_2S$, the decomposition tests were first performed galvanostatically at a current of 0.112 mA until the potential was low than 1.7 V. Then the cells were potentiostatically charged at 2.35 V until the current was down to 10$^{-5}$ A. The whole $Li_2S$ nucleation/ decomposition tests were carried out on an Ivium Vertex.One Electrochemical Workstation.

### Assembly of symmetric cells

Sulfur and $Li_2S$ at a molar ratio of 5:1 were dissolved in a mixture of 1,2-dimethoxyethane (DME)/1,3-dioxolane (DOL) solution containing 1.0 mol L$^{-1}$ LiTFSI and 2 wt% LiNO$_3$ to prepare $Li_2S_6$ electrolyte. Symmetric cells were assembled by applying CP/Cu-2 as the working and counter electrodes, and the $Li_2S_6$ electrolyte usage was 20 μL. CV tests were conducted between −1 and 1 V at the scan rates of 50 and 0.5 mV s$^{-1}$. CV profiles of the symmetric cells were collected on a Metrohm Autolab G204 Electrochemical Workstation.

### Li plating/stripping tests

Cu//Li cells were assembled by using Li foil as the anode, Cu foil as the cathode with different catalyst coated separators, and 20 μL mixture of DME/DOL solution with 1.0 mol L$^{-1}$ LiTFSI and 2.0 wt% LiNO$_3$ as the electrolyte. Cu//Li cells were discharged at a current density of 1.0 mA cm$^{-2}$ and a capacity of 1.0 mA h cm$^{-2}$, then striped at 1.0 mA cm$^{-2}$ on a Neware Battery Measurement System. And the Li//Li cells were assembled with two Li foils as the working electrode, plating and stripping at the needed current. And the Cu foils and Li foils were purchased from Canrd Technology Co. Ltd (China).

### Electrochemical evaluations

The slurry was obtained by mixing S/electrocatalyst composite, Super P, and LA133 aqueous binder with a mass ratio of 8:1:1. The as-achieved slurry was subsequently coated on a piece of Al foil and followed by vacuum drying at 60 °C for 12 h. The cathode was cut into circular disks with a diameter of 13 mm prior to use. The sulfur content and mass loading were 60 wt%, 1.4–1.6 mg cm$^{-2}$, respectively. The coin-type batteries were assembled with Celgard 2500 PP membrane as the separator, Li foil as the anode, and mixture of DME/DOL solution containing 1.0 mol L$^{-1}$ LiTFSI and 2 wt% LiNO$_3$ as the electrolyte. The electrolyte/sulfur ratio was 15.0 and 4.8 μL mg$^{-1}$ for routine and high-load batteries, respectively. Galvanostatic discharge/charge, rate, and cycling performance measurements were carried out on a Neware Battery Measurement System in the voltage window of 2.8–1.7 V. CV, LSV, and EIS curves were recorded on a Metrohm Autolab G204 Electrochemical Workstation. Tafel plots of were obtained by the two peaks from LSV curves with the detected the voltage window of 2.8–1.7 V. The iR-compensation isn't performed because of the relatively small values. And all tests were performed at 28 °C in a thermostatic test chamber.

### Synchrotron radiation X-ray 3D nano-CT tests

The synchrotron radiation X-ray 3D nano-CT was carried out on the beamline BL07W at the National Synchrotron Radiation Laboratory (NSRL), Hefei, China, (flux: $2 \times 10^{10}$ Phs s$^{-1}$, spatial resolution: 30 nm). In details, the cells after $Li_2S$ nucleation tests were disassembled to achieve the powder sample. Under pure argon atmosphere, the samples were dropped onto a piece of carbon-free formvar film with 100 mesh, and dried at room temperature. The loaded formvar film was further transferred into the vacuum chamber for imaging tests. Two-dimensional (2D) tomograms were collected at a title-angle range of

−60° to 60° with an interval of 1°. The exposure time was 2 s. The obtained 2D tomograms were reconstructed into 3D visual images by virtue of using X Radia XMR reconstructor software. Further 3D visualization and segmentation of the reconstructed images were performed on an Avizo Fire VSG software (Visualization Sciences Group, Bordeaux).

### Small angle neutron scattering tests

The samples were collected from the cycled Li//Li symmetric cells. After washing with DME solution and naturally drying in argon-filled glove box, the lithium anode was sealed into Al-plastic film for SANS analysis on the SANS-Suanni spectrometer at China Mianyang Research Reactor. The neutron wavelength used in this experiment was 0.53 nm, and 1.5 and 9.9 m of three sample-to-detector distance were selected, respectively. BerSANS software was applied to convert the original data to one-dimension (1D) data.

### Reporting summary

Further information on research design is available in the Nature Portfolio Reporting Summary linked to this article.

## Data availability

The data supporting the findings of this work are available within the article and its Supplementary Information files. All other relevant data supporting the findings of this study are available from the corresponding author on request. The single crystal structure data have been deposited in the Cambridge Crystallographic Data Centre (CCDC Nos. 2217502 and 2047772) (https://www.ccdc.cam.ac.uk/). Source data are provided with this paper.

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

## Acknowledgements

The investigation was supported by the National Natural Science Foundation of China (Grant No. 52172239), Project of State Key Laboratory of Environment-Friendly Energy Materials (Grant Nos. 21fksy24 and 18ZD320304) and Frontier Project of Chengdu Tianfu New Area Institute (SWUST, 2022ZY017). The supports from the State Key Laboratory of Environment-Friendly Energy Materials (Mianyang, China) and the beamline BL07W at National Synchrotron Radiation Laboratory (Hefei, China) are acknowledged.

## Author contributions

Y.S. and G.Z. conceived and designed the experiments. Q.Y. prepared the electrode materials and conducted the characterizations of materials and LSBs. J.C. performed the theoretical simulations with the G.W.'s guidance. G.L. carried out the finite element simulations with the guidance of D.W. and W.Z., R.G., Z.H. and Z.S. helped to analyze the working mechanism of catalysts. J.H. performed the SANS tests with D.L.'s guidance. L.S. conducted the synchrotron radiation X-ray 3D nano-CT tests. Q.Y. and Y.S. were mainly responsible for preparing the manuscript with input from all other authors. All authors discussed the results and commented on the manuscript. All authors have given approval to the final version of the manuscript.

## Competing interests

The authors declare no competing interests.
