## [Peer Review File · Nature Communications]

Chlorine Bridge Bond-Enabled Binuclear Copper Complex for Electrocatalyzing Lithium–Sulfur ReactionsREVIEWER COMMENTS

Reviewer #1 (Remarks to the Author):

In this work, the authors describe the synthesis and testing of a homonuclear Cu dual-atom catalyst which is able to facilitate sulfur to lithium sulfide redox conversion as the electrode and regulate Li deposition behaviour when employed as a separator coating. The electrochemical results are impressive, and Cu-2 undoubtedly outperforms the Cu-1 and rphenGO control samples, but I think the authors do not present sufficient evidence to support their assertion that it is a synergistic effect of the Cu dual atom site that supports this performance.

Although the characterisation techniques employed are comprehensive, the work would benefit from additional clarification on the following points:

1. In the HAADF-STEM images, it is not obvious that the circled dual- or single-atom sites in d and f respectively are statistically significant. Do the authors have images of additional areas of the sample?

2. Operando Raman: Could the authors clarify the configuration of the operando Raman cell? Was a hole cut through the cathode/anode? Was the measurement taken on the cathode/anode side? Would the authors expect to see different variations in the signals of individual soluble polysulfides as they interconvert rather than a uniform decrease then increase during discharge/charge? A comparison with the Cu-1 and rphenGO materials would also be useful here.

3. In the Gibbs free energy profiles for polysulfide conversion reactions, could the authors comment on the significance that the rate determining step is different for Cu-1 (Li₂S₄ to Li₂S₂) than Cu-2 and rphenGO (Li₂S₂ to Li₂S)?

4. In the potentiostatic tests, why is the charge capacity obtained so much higher than the discharge capacity? Can the authors comment on the electrocatalytic ability of the Cu sites in both the discharge and charge direction?

5. In the experimental methods section, it is not clear to me whether Cu-2 contains twice the amount of Cu as Cu-1 or half the amount of phenanthroline in the overall material. It seems to me that if the amount of Cu is double then the electrochemical performance should obviously be improved. Can the authors comment/clarify? Also, I think the authors mean phen:Cu 2:1 rather than 1:1 as written in the methods section. Why was n-butanol used for the preparation of Cu-1 and ethanol or Cu-2?

6. For the tests on the separator coated with catalyst, is there a risk that the slurry fully penetrates the separator leading to short circuit in the cell? Short circuit behaviour can sometimes look like perfect Li plating behaviour.

7. For the slurry cast electrodes, can the authors suggest why more cracking might be observed in the rphenGO and rGO electrodes than the Cu-1 and Cu-2 materials?

8. Why were slurry cast electrodes used for electrochemical testing and carbon paper coated electrodes used for the other tests?

9. The manuscript would benefit from additional proofreading.

Overall, while I think the authors have conducted an in-depth study, they do not present compelling enough justification for the claim that synergistic behaviour at the Cu dual-atom site is the reason for superior performance of Cu-2 over Cu-1.

Reviewer #2 (Remarks to the Author):

Summary:

This Research paper NCOMMS-23-54974 titled, "Chlorine Bridge Bond-Enabled Binuclear Copper Complex for Electrocatalyzing Lithium-Sulfur Reactions," reports a homonuclear Cu dual-atom catalyst (referred to as Cu-2) that is introduced for high-performance lithium-sulfur cells. The catalyst features a proximal distance of 3.5 Å between two adjacent Cu atoms, connected by symmetrical chlorine bridge bonds. The unique coordination of Cu-2 with Cl and N atoms, along with the synergistic effect of proximal Cu atoms, enhances the active interface concentration. This is observed through synchrotron radiation X-ray 3D nano-computed tomography and small-angle neutron scattering results, which reveal the synchronized evolution behaviors of sulfur and lithium species. The design of Cu-2 overcomes the activity limitations of mononuclear metal centers, presenting a novel catalyst concept for lithium-sulfur cells. The resulting S/Cu-2 cathode exhibits an initial capacity of 1140.6 mA h g⁻¹ at a C/5 rate and makes a high sulfur loading of 7.7 mg cm⁻² and an electrolyte usage of 4.8 μL mgS⁻¹ in cell an areal capacity of 7.8 mA h cm⁻².

General comment:

This study reports on the cathode design and preparation for the lithium-sulfur cells. The optimal cathode is used to show the improved battery performance along with the support of material, electrochemical and cell analysis. The overall quality is good. Before the acceptance, some revision is suggested, and I hope the authors find the comment useful.

Comments:

(1) In the main text, the Methods section is suggested to include the facilities and parameters for the electrochemical analysis. It is also necessary to report the lithium-sulfur cell preparation and testing parameters in the main text, especially the sulfur loading and content as well as the amount of electrolyte that highlighted in the abstract.

[Suggestion] Please report the important information in the main text.

(2) In the materials analysis, the UV-vis data is suggested to support with its y axis to know the data analysis. In the FTIR analysis, in addition to the shift of C=N and C=C bonds, the spectra seem to show that all peaks are shifted after coordinating with Cu²⁺. It is suggested to explain this.

[Suggestion] Please check the data in UV-vis and FTIR analysis.

(3) In the electrochemical analysis, the galvanostatic discharge-charge curves of different cycles at 0.2 C for S/Cu-2 and S/Cu-1 is necessary to add the charge curve to complete the data. The rate performance is suggested to add the discharge/charge efficiency. The "C" and "capacity" reported in plots and main text are suggested to be clarified as discharge capacity.

[Suggestion] Please add the missing data to the electrochemical analysis.

(4) The electrochemical analysis and the cell performance are obtained from two different cells. The high-loading sulfur cathode in the cell with low amount of electrolyte seem to suffer a low cycling rate and low electrochemical utilization of sulfur in a short cycle life. The development of lithium-sulfur cells with a high amount of sulfur and low amount of electrolyte is important. It is suggested to provide some discussion on this. It is also suggested to include the recent work in making lean-electrolyte lithium-sulfur cell with a high-loading cathode.

[Suggestion] Please make a discussion and reference discussion on the sulfur loading and electrolyte amount.

(5) Chlorine bridge bond-enabled binuclear copper complex is reported for electrocatalyzing lithium-sulfur battery reaction. Since the electrocatalysis reaction is highlighted in the topic, it is suggested to make a discussion on the electrocatalysis reaction by explaining the reaction intermediate and its

active energy toward (or compared to) the lithium-sulfur reaction.
[Suggestion] Please give more information on the electrocatalysis reaction.

Reviewer #1 (Remarks to the Author):

In this work, the authors describe the synthesis and testing of a homonuclear Cu dual-atom catalyst which is able to facilitate sulfur to lithium sulfide redox conversion as the electrode and regulate Li deposition behaviour when employed as a separator coating. The electrochemical results are impressive, and Cu-2 undoubtedly outperforms the Cu-1 and rphenGO control samples, but I think the authors do not present sufficient evidence to support their assertion that it is a synergistic effect of the Cu dual atom site that supports this performance.

Although the characterisation techniques employed are comprehensive, the work would benefit from additional clarification on the following points:

Our response:

We thank the reviewer for these positive and helpful comments. Our response follows each comment. Additionally, we agree with the reviewer that the currently obtained experimental and theoretical results don't sufficiently support the expression of "synergistic effect of the Cu dual atom sites". Accordingly, we have revised all the expressions of "synergistic effect" in the manuscript to well support the electrochemical performance of batteries.

1. In the HAADF-STEM images, it is not obvious that the circled dual- or single-atom sites in d and f respectively are statistically significant. Do the authors have images of additional areas of the sample?

Our response:

We are appreciative for this comment. We provide HAADF-STEM images of additional areas of samples (Figure S5a and S5c). We also circled the proximal two Cu atoms in the HAADF-STEM images of samples and calculated the distances between them by using the Digital Micrograph software (Figure S5b and S5d). As seen in Figure S5b, the distances between the proximal two Cu atoms in the HAADF-STEM image of Cu-2 are around 0.35 nm, in accordance with the results of single crystal XRD tests. For the Cu-1 sample, the calculated distances between the neighboring two Cu atoms are obviously fluctuant (Figure S5d). The above results demonstrate that the Cl-bridge pairs maintain the structural stability of proximal two Cu atom in Cu-2 sample.

“The distances between the proximal two Cu atoms in the HAADF-STEM image of Cu-2 are around 0.35 nm, in accordance with the results of single crystal XRD tests (Figure S5a and S5b). For the Cu-1 sample, the calculated distances between the neighboring two Cu atoms are obviously fluctuant (Figure S5c and S5d). The above results demonstrate that the Cl-bridge pairs maintain the structural stability of proximal two Cu atoms in Cu-2 sample.”

Figure S5. HAADF-STEM images of Cu-2 (a) and Cu-1 (c). The calculated distances of the neighboring two Cu atoms in the HAADF-STEM images of Cu-2 (b) and Cu-1 (d).

2. Operando Raman: *Could the authors clarify the configuration of the operando Raman cell? Was a hole cut through the cathode/anode? Was the measurement taken on the cathode/anode side? Would the authors expect to see different variations in the signals of individual soluble polysulfides as they interconvert rather than a uniform decrease then increase during discharge/charge? A comparison with the Cu-1 and rphenGO materials would also be useful here.*

Our response:

We thank the reviewer for the professional comment. The *operando* Raman cell device was purchased from Beijing Scistar Technology Co., Ltd. The components of the *operando* Raman cell device were illustrated in Figure R1 and the Raman measurements were only taken on the cathode side. Typically, a piece of Al mesh with a pore size of 0.5 mm × 0.1 mm was used as the current. The cathode was obtained by the disconnected coverage of the slurry on the Al mesh. After drying, the cathode was cut into small disks with a diameter of 13.0 mm for the *operando* Raman cell fabrication. Note that the disconnected material coverage simultaneously provides the cathode and catholyte regions for the *operando* Raman signal collection. Such an *operando* Raman cell assembling route has been also reported by our previous investigations (*Adv. Sci.* **2022**, 9, 2204027; *Nano Energy* **2021**, 89, 106414). The comparison of Raman signals with regards to different samples can be depicted their ability for suppressing shuttle effect. Generally, the catholyte region more clearly reflect the Raman signals of soluble polysulfides than the cathode region owing to the existence of sulfur, carbon and binder in cathode region. Accordingly, we provided the *operando* Raman spectra of catholyte regions with respects to Cu-1 and rphenGO, as displayed in Figure S14.

“The cathode was obtained by the disconnected coverage of the slurry on a piece of Al mesh with a pore size of 0.5 mm × 0.1 mm. After drying, the cathode was cut into small disks with a diameter

of 13.0 mm for the *operando* Raman cell fabrication. Note that the disconnected material coverage simultaneously provides the cathode and catholyte regions for the *operando* Raman signal collection.”

“*Operando* Raman spectra of catholyte regions with respects to Cu-1 and rphenGO were also shown in Figure S14. The catholyte region of S/Cu-2 shows the weaker soluble LiPS Raman signals along with the discharge proceeding in contrast to the other two samples. Particularly, the Raman signals of LiPSs in catholyte region of S/Cu-2 almost disappear at the end of discharge. These results further confirm the stronger LiPS anchoring ability of Cu-2. ^{50,51}”

“50. Shi, Z., Sun, Z., Cai, J., Fan, Z., Jin, J., Wang, M., Sun, J. Boosting Dual-Directional Polysulfide Electrocatalysis *via* Bimetallic Alloying for Printable Li–S Batteries. *Adv. Funct. Mater.* **2021**, 31, 2006798.

51. Wang, M., Sun, Z., Ci, H., Shi, Z., Shen, L., Wei, C., Ding, Y., Yang, X., Sun, J. Identifying the Evolution of Se-Vacancy-Modulated MoSe₂ Pre-Catalyst in Li–S Chemistry. *Angew. Chem. Int. Ed.* **2021**, 60, 24558.”

Figure R1. Components of the *operando* Raman cell device.

Figure S14. Operando Raman spectra of the catholyte regions with regards to S/rPhenGO, S/Cu-1 and S/Cu-2 during the discharge/charge process at 0.2 C.

3. In the Gibbs free energy profiles for polysulfide conversion reactions, could the authors comment on the significance that the rate determining step is different for Cu-1 (Li_2S_4 to Li_2S_2) than Cu-2 and rphenGO (Li_2S_2 to Li_2S)?

Our response:

We are grateful for the reviewer's instructive comment. Based on the DFT results, the negative Gibbs free energy change (ΔG) values present thermodynamically spontaneous exothermic steps and the positive values are the thermodynamically endothermic steps. The rate-limiting step in the

entire sulfur reduction process is determined by the biggest positive ΔG value. And such rate determining step is closely related to the bond strength between catalyst and sulfur species. Along this line, the configurations of catalyst and sulfur species have a critical effect on the bond strength, leading to the different biggest positive ΔG values, implying the distinct rate determining step for Cu-1 (Li_2S_4 to Li_2S_2) and Cu-2 (Li_2S_2 to Li_2S). Such a phenomenon has been also reported by the previous literatures (*Nat. Nanotechnol.* **2021**, 16, 166; *ACS Nano* **2022**, 16, 6414).

4. In the potentiostatic tests, why is the charge capacity obtained so much higher than the discharge capacity? Can the authors comment on the electrocatalytic ability of the Cu sites in both the discharge and charge direction?

Our response:

Thanks for the referee's professional comment. The potentiostatic tests were conducted based on the Li_2S nucleation and decomposition procedures. In principle, the calculated Li_2S precipitation capacity pertains to the contribution of the Li_2S_4 (by subtracting the contributions of Li_2S_8 and Li_2S_6 in the total capacity) (*Angew. Chem. Int. Ed.* **2022**, 61, 2204327; *Adv. Energy Mater.* **2020**, 10, 2002271; *ACS Nano* **2021**, 15, 13436). The Li_2S decomposition capacity is attained by calculating the contribution of all the Li_2S deposit originating from the conversion of Li_2S_8 , Li_2S_6 and Li_2S_4 in the nucleation procedure. Therefore, the Li_2S decomposition capacity is much higher than the Li_2S precipitation capacity (*Adv. Mater.* **2020**, 32, 2000315; *ACS Energy Lett.* **2020**, 5, 3041; *Energy Storage Mater.* **2022**, 49, 153). In addition, we have added the comment on the electrocatalytic ability of the Cu sites in both the discharge and charge direction.

“Note that the higher activity for the Li–S redox reactions is harvested by Cu-2 than other samples owing to the identified homonuclear dual Cu centers according to the above potentiostatic test results. The proximal binuclear Cu centers in Cu-2 as well as their optimized coordination environment provide more active interfaces for realizing the appropriate electrokinetic control of the Li_2S nucleation and decomposition reactions, manifesting the kinetically promoted discharge and charge processes of the practically operated LSBs.”

5. In the experimental methods section, it is not clear to me whether Cu-2 contains twice the amount of Cu as Cu-1 or half the amount of phenanthroline in the overall material. It seems to me that if the amount of Cu is double then the electrochemical performance should obviously be improved. Can the authors comment/clarify? Also, I think the authors mean phen:Cu 2:1 rather than 1:1 as written in the methods section. Why was n-butanol used for the preparation of Cu-1 and ethanol or Cu-2?

Our response:

We are grateful for the reviewer's valuable comment. The 1-Cuphen for preparing Cu-1 was synthesized with the mole ratio of copper (II) chloride dihydrate and 1,10-phenanthroline as 1:2. And the 2-Cuphen for preparing Cu-2 was synthesized with the mole ratio of copper (II) chloride dihydrate and 1,10-phenanthroline as 1:1. The amount of Cu centers in Cu-1 is the same as that in Cu-2 owing to the consistent usage of copper (II) chloride dehydrate theoretically. Moreover, we used the ICP-OES to experimentally probe the contents of Cu centers in Cu-1 and Cu-2. The ICP-OES measurement results show that the Cu contents in Cu-1 and Cu-2 are 1.16 and 1.18 wt.%, respectively, substantiating the same contents of Cu centers in the two catalysts. According to our exploration results, 1-Cuphen presents superior solubility in ethanol, resulting in the practical difficulty to precipitate the 1-Cuphen single crystal, which is thus unfavorable for the subsequent single crystal XRD tests. Therefore, the appropriate solvents are selected for 1-Cuphen and 2-Cuphen for successfully precipitating the single crystals.

“Inductively Coupled Plasma Optical Emission Spectrometer (ICP-OES) were conducted to experimentally probe the content of Cu centers in Cu-1 and Cu-2. According to the measurement results, the Cu contents in Cu-1 and Cu-2 are 1.16 and 1.18 wt.%, respectively, substantiating the same contents of Cu centers in the two catalysts.”

6. For the tests on the separator coated with catalyst, is there a risk that the slurry fully penetrates the separator leading to short circuit in the cell? Short circuit behaviour can sometimes look like perfect Li plating behaviour.

Our response:

We thank the reviewer for the constructive comment. We have provided the digital graphs of the coated separator with the catalyst slurry, as displayed in Figure R2. The separator side toward the cathode is fully covered with the Cu-2 and binder. And the separator toward the anode is pure in color and no catalyst penetrates the separator. In addition, we also impaled the separator by using a pin and tested the voltages of the batteries with bare, Cu-2 loaded and impaled separator. It can be seen from Figure R2, the batteries with the bare and Cu-2 loaded separator show the normal voltages of around 2.8 V. And the battery with the impaled Cu-2 loaded separator only displays the voltage of around 2.3 V, implying the occurred phenomenon of short circuit. The above results demonstrate that the covered catalyst cannot penetrate the separator and thus cause the short circuit of battery. Such fact has been also reported by previous investigations (*Adv. Mater.* **2022**, 34, 2107638; *Adv. Funct. Mater.* **2022**, 32, 2204635; *ACS Nano* **2021**, 15, 13436). The EIS profiles of cycled short-circuit Li//Li cell and Li//Li cell with Cu-2 coated separator in Figure R3. In Figure R3a, the observed abnormal curve shape is attained by the Li//Li symmetric cell owing to the

occurred short circuit phenomenon (*Joule* **2022**, 6, 273). And the EIS profiles of the Li//Li symmetric cells with Cu-2 based separator in Figure R3b present the typical interfacial charge transfer behavior, confirming no short circuit phenomenon occurring in the cell.

Figure R2. Digital graphs of bare PP (a, b), Cu-2/PP (d, e) and impaled Cu-2/PP separator (g, h); voltage tests of the assembled batteries with the corresponding separators.

Figure R3. EIS curves of short-circuit Li//Li cell (a) and Li//Li cell with Cu-2 coated separator (b) after working for 100 cycles.

7. For the slurry cast electrodes, can the authors suggest why more cracking might be observed in the rphenGO and rGO electrodes than the Cu-1 and Cu-2 materials?

Our response:

We are grateful for the instructive comment. Sulfur cathodes produce the cracks during the long-term cycling owing to the density difference between sulfur and the discharge product of Li_2S . The employment of highly active catalysts can rationalize the conversion reactions between sulfur and Li_2S , and the possible reasons can be conducted as follows: (i) the sulfur is restrained in the cathode

side and the Li_2S deposition on the surface of anode is effectively avoided by the Cu-2, leading to the superior structural integrity of the sulfur cathode (*Adv. Mater.* **2022**, 34, 2202256; *Nano Lett.* **2022**, 22, 3728); (ii) the homogeneous spatial distribution and small size of Li_2S resulting from the catalytic effect of the Cu-2 avoid the structure hollowing out largely and benefit to the stress transferring uniformly in the cathode (*Adv. Funct. Mater.* **2021**, 31, 2101285; *Adv. Sci.* **2022**, 9, 2204027); (iii) the Cu-2 catalyst is favorable for building a strong SEI film, which is thereby beneficial to dictate a better morphology of the sulfur cathode (*Adv. Funct. Mater.* **2023**, 33, 2306321). Therefore, the highly active catalyst can help to maintain the structural integrity of sulfur cathode and thus shows the durable performance, which has been also confirmed by other literatures (*Adv. Mater.* **2016**, 28, 9551; *Adv. Funct. Mater.* **2021**, 31, 2100793)

8. Why were slurry cast electrodes used for electrochemical testing and carbon paper coated electrodes used for the other tests?

Our response:

We are appreciated for the valuable comment. In this manuscript, the slurry cast electrodes were assembled for rate and cycling performance evaluation, and the carbon paper-based electrodes were used for tests of Li_2S nucleation/decomposition and CV curves of Li_2S_6 symmetric cells. The slurry cast electrodes were prepared for performance tests based on the industrial production technologies of batteries. The use of commercial carbon paper aims to accurately evaluate the activity of catalysts, and the detailed reasons involves the following aspects: (i) the carbon paper provides 3D frameworks to spatially support catalyst, pledging the sufficient contact of catalyst and sulfur species; (ii) the carbon paper shows nonpolar surface, benefiting to the accurately identify the chemical affinity of sulfur species by catalyst; (iii) the self-supporting design of carbon paper avoid the use of binder, which is favorable for eliminating the interference. Such approaches of electrode preparation for electrochemical measurements have been extensively applied by the previous investigations (*Angew. Chem. Int. Ed.* **2021**, 60, 24558; *Adv. Mater.* **2021**, 33, 2103050; *Adv. Mater.* **2018**, 30, 1705219).

9. The manuscript would benefit from additional proofreading.

Overall, while I think the authors have conducted an in-depth study, they do not present compelling enough justification for the claim that synergistic behaviour at the Cu dual-atom site is the reason for superior performance of Cu-2 over Cu-1.

Our response:

We thank the reviewer for the constructive comment. We have proofread the manuscript and revised some small grammar errors. In addition, we agree with the reviewer that the present experimental and theoretical results don't sufficiently support the conclusion of "synergistic effect

of the Cu dual atom sites”. Therefore, we have revised all the expressions of “synergistic effect” in the manuscript.

Reviewer #2 (Remarks to the Author):

Summary: *This Research paper NCOMMS-23-54974 titled, “Chlorine Bridge Bond-Enabled Binuclear Copper Complex for Electrocatalyzing Lithium–Sulfur Reactions,” reports a homonuclear Cu dual-atom catalyst (referred to as Cu-2) that is introduced for high-performance lithium-sulfur cells. The catalyst features a proximal distance of 3.5 Å between two adjacent Cu atoms, connected by symmetrical chlorine bridge bonds. The unique coordination of Cu-2 with Cl and N atoms, along with the synergistic effect of proximal Cu atoms, enhances the active interface concentration. This is observed through synchrotron radiation X-ray 3D nano-computed tomography and small-angle neutron scattering results, which reveal the synchronized evolution behaviors of sulfur and lithium species. The design of Cu-2 overcomes the activity limitations of mononuclear metal centers, presenting a novel catalyst concept for lithium-sulfur cells. The resulting S/Cu-2 cathode exhibits an initial capacity of 1140.6 mA h g⁻¹ at a C/5 rate and makes a high sulfur loading of 7.7 mg cm⁻² and an electrolyte usage of 4.8 μL mg^s⁻¹ in cell an areal capacity of 7.8 mA h cm⁻².*

General comment:

This study reports on the cathode design and preparation for the lithium-sulfur cells. The optimal cathode is used to show the improved battery performance along with the support of material, electrochemical and cell analysis. The overall quality is good. Before the acceptance, some revision is suggested, and I hope the authors find the comment useful.

Our response:

We thank the reviewer for these positive and helpful remarks. Our response follows each comment.

1. *In the main text, the Methods section is suggested to include the facilities and parameters for the electrochemical analysis. It is also necessary to report the lithium-sulfur cell preparation and testing parameters in the main text, especially the sulfur loading and content as well as the amount of electrolyte that highlighted in the abstract.*

Our response:

We thank reviewer for the professional comment. We have provided the facilities and parameters for the electrochemical analysis. And lithium-sulfur cell preparation and testing parameters in the

main text, especially the sulfur loading and content as well as the amount of electrolyte that highlighted in the abstract are also added into the revised manuscript.

“Li₂S nucleation/ decomposition tests

0.2 mol L⁻¹ Li₂S₈ solution was first attained by mixing sulfur and Li₂S at a molar ratio of 7:1 in tetraglyme solvent. CP was punched into small disks with a diameter of 13 mm. Cu-1, Cu-2 and rphenGO powder were dispersed into ethanol and then loaded on CP. Cells were fabricated by employing CP-Cu-1 or CP-Cu-2 as the cathode and lithium foil as the anode, 20 μL Li₂S₈ solution as the catholyte and 20 μL LiTFSI (1.0 mol L⁻¹) solution without Li₂S₈ as the anolyte. The assembled cells were first galvanostatically discharged to 2.06 V at a current of 0.112 mA and then discharged potentiostatically at 2.05 V until the current was below 10⁻⁵ A. The nucleation capacity of Li₂S was calculated by the integral area of the plotted curve based on Faraday's Law. For the Li₂S decomposition tests, the cells were fabricated by following the same assemble process as the nucleation test. To ensure the complete transformation of LiPSs into Li₂S, the decomposition tests were first performed galvanostatically at a current of 0.112 mA until the potential was low than 1.7 V. Then the cells were potentiostatically charged at 2.35 V until the current was down to 10⁻⁵ A. The whole Li₂S nucleation/ decomposition tests were carried out on an Ivium Vertex.One Electrochemical Workstation.”

“Assembly of symmetric cells

Sulfur and Li₂S at a molar ratio of 5:1 were dissolved in a mixture of 1,2-dimethoxyethane (DME) /1,3-dioxolane (DOL) solution containing 1.0 mol L⁻¹ LiTFSI and 2 wt.% LiNO₃ to prepare Li₂S₆ electrolyte. Symmetric cells were assembled by applying CP-Cu-2 as the working and counter electrodes, and the Li₂S₆ electrolyte usage was 20 μL. Cyclic voltammetry (CV) tests were conducted between -1 to 1 V at the scan rate of 50 and 0.5 mV s⁻¹. The CV profiles of the symmetric cells were collected on a Metrohm Autolab G204 Electrochemical Workstation.”

“Li plating/stripping tests

Cu//Li cells were assembled by using Li foil as the anode, Cu foil as the cathode and different catalyst coated separators, and 20 μL mixture of DME/DOL solution with 1.0 mol L⁻¹ LiTFSI and 2.0 wt.% LiNO₃ as electrolyte. Cu//Li cells were discharged at a current density of 1.0 mA cm⁻² and a capacity of 1.0 mA h cm⁻², then striped at 1.0 mA cm⁻² on a Neware Battery Measurement System. And the Li//Li cells were assembled with two Li foils as the working electrode, plating and stripping at the needed current.”

“Electrochemical evaluations

The slurry was obtained by mixing S/electrocatalyst composite, Super P and LA133 aqueous binder with a mass ratio of 8:1:1. The as-achieved slurry was subsequently coated on a piece of Al foil and followed by vacuum drying at 60 °C for 12 h. The cathode was cut into circular disks with a diameter of 13 mm prior to use. The sulfur content and mass loading were 60 wt.%, 1.4–1.6 mg cm⁻², respectively. The content of electrocatalyst can be adjusted into 10 and 20 wt.%. The coin-type batteries were assembled with Celgard 2500 PP membrane as the separator, Li foil as the anode and mixture of DME/ DOL solution containing 1.0 mol L⁻¹ LiTFSI and 2 wt.% LiNO₃ as the electrolyte. The electrolyte/sulfur ratio was 15.0 and 4.8 μL mg⁻¹ for routine and high-load batteries, respectively. Galvanostatic discharge/charge, rate and cycling performance measurements were carried out on a Neware Battery Measurement System in the voltage window of 2.8–1.7 V. CV, LSV and EIS curves were recorded on a Metrohm Autolab G204 Electrochemical Workstation.”

“As a result, thanks to the proximity effect, the as-derived S/Cu-2 cathode obtains a remarkable initial capacity of 1140.6 mA h g⁻¹ at 0.2 C with negligible decay. Equally importantly, a remarkable areal capacity of 7.8 mA h cm⁻² is achieved under the scenario of sulfur content of 60 wt.%, sulfur mass loading of 7.7 mg cm⁻² and electrolyte dosage of 4.8 μL mg⁻¹.”

2. In the materials analysis, the UV-vis data is suggested to support with its y axis to know the data analysis. In the FTIR analysis, in addition to the shift of C=N and C=C bonds, the spectra seem to show that all peaks are shifted after coordinating with Cu²⁺. It is suggested to explain this.

[Suggestion] Please check the data in UV-vis and FTIR analysis.

Our response:

We are grateful for the reviewer’s meaningful comment. The Y axis information for UV–vis data has been provided, as displayed in Figure 2a. The shift of all peaks in complexes has been also discussed in the revised manuscript. The formation of Cu–N bonds in complexes tends to change the configuration, symmetry and electron density distribution of the whole ligand, hence leading to the shift of all the peaks. Such a phenomenon has been also reported by previous literatures (*J. Am. Chem. Soc.* **2006**, 128, 16515; *J. Am. Chem. Soc.* **2011**, 133, 10081; *Dalton Trans.* **2022**, 51, 6314).

“Note that the formation of Cu–N bonds in complexes tends to change the configuration, symmetry and electron density distribution of the whole ligand, hence leading to the shift of all the peaks.”

Figure 2. (a) UV-vis spectra of phen, 1-Cuphen and 2-Cuphen.

3. In the electrochemical analysis, the galvanostatic discharge-charge curves of different cycles at 0.2 C for S/Cu-2 and S/Cu-1 is necessary to add the charge curve to complete the data. The rate performance is suggested to add the discharge/charge efficiency. The “C” and “capacity” reported in plots and main text are suggested to be clarified as discharge capacity.

Our response:

We thank the reviewer for the valuable suggestions. Galvanostatic charge curves with different cycles at 0.2 C for S/Cu-2 and S/Cu-1 have been added in Figure 6d and 6e. The Coulombic efficiencies for the rate performance have been also provided in Figure 6a. Following the reviewer’s suggestion, “C” and “capacity” have been clarified as “Discharge capacity”.

“On the proof of this concept, we introduce chlorine (Cl) bridge bond-enabled atomically dispersed Cu-based complexes as the electrocatalysts in LSBs and investigate the battery performance, involving discharge capacity (C), rate capability as well as cycling stability.”

“Even with a high sulfur loading of 7.7 mg cm^{-2} , the S/Cu-2 cathode can obtain an areal capacity (C_A) of 7.8 mA h cm^{-2} , which propels the real implementation of highly efficient and remarkably durable LSBs”

Figure 6. Galvanostatic discharge and charge curves of S/Cu-2 (d) and S/Cu-1 (e).

Figure 6. (a) Rate capabilities of S/Cu-1, S/Cu-2, S/rphenGO and S/rGO cathodes.

4. The electrochemical analysis and the cell performance are obtained from two different cells. The high-loading sulfur cathode in the cell with low amount of electrolyte seem to suffer a low cycling rate and low electrochemical utilization of sulfur in a short cycle life. The development of lithium-sulfur cells with a high amount of sulfur and low amount of electrolyte is important. It is suggested to provide some discussion on this. It is also suggested to include the recent work in making lean-electrolyte lithium-sulfur cell with a high-loading cathode. [Suggestion] Please make a discussion and reference discussion on the sulfur loading and electrolyte amount.

Our response:

We appreciate for the reviewer's professional comment. The high sulfur loading and low electrolyte usage are indeed of utmost importance to practically implementation of LSBs. Along this line, we provide the cycling performance of batteries with the high sulfur loading of 7.7 mg cm^{-2} and low electrolyte usage of $4.8 \text{ } \mu\text{L mg}^{-1}$ at 0.1 C. The battery cycling performance comparison between this work and other reports based on high sulfur loadings and low electrolyte usage has been also listed in a new Table S3. As seen from Table S3, the battery with adding Cu-2 harvests the favorable areal capacity of 7.8 mA h cm^{-2} and a capacity retention of 84.6% over 50 cycles at 0.1 C under the conditions of sulfur loading of 7.7 mg cm^{-2} and electrolyte usage of $4.8 \text{ } \mu\text{L mg}^{-1}$. Note that the cycling lifespan of the high-loaded cathode need to be further optimized towards commercially viable LSBs throughout unremitting endeavor from the comprehensive design of catalysts, electrolytes, binders, *etc.* And such explorations of key materials for LSBs are also ongoing in our laboratory.

“As displayed in Figure 6g and S31, with a sulfur loading of 5.1 and 7.7 mg cm^{-2} , the S/Cu-2 cathode can respectively achieve the superior initial areal capacities of 5.1 and 7.8 mA h cm^{-2} , and

retentions of 91.2% and 84.6% at 0.1 C over 50 cycles, verifying the remarkable catalytic effect of Cu-2. The S/Cu-2 cathode with the sulfur loading of 7.7 mg cm⁻² exhibits more favorable electrochemical performance in contrast to the previously reported high-loaded cathodes (Figure 6h and Table S3). The high sulfur loading and low electrolyte usage are of utmost importance to the practically implementation of LSBs. Therefore, the cycling lifespan of the high-loaded cathodes need be further optimized towards commercially viable LSBs throughout unremitting endeavor from the comprehensive design of catalysts, electrolytes, binders, *etc.*”

Table S3. Electrochemical performance comparison of high-sulfur-load batteries between this work and recent literatures.

Sulfur loading (mg cm ⁻²)	Areal capacity (mA h cm ⁻²)	Capacity retention (%)	Cycle number	Rate (C)	E/S (μL mg ⁻¹)	Ref.
7.7	7.8	84.6	50	0.1	4.8	This work
4.9	4.7	85.1	100	0.1	-	1
5.8	5.0	88.0	50	0.1	-	2
5.3	5.3	83.1	40	0.2	7.5	3
5.1	5.0	80.0	50	0.1	10.0	4
5	4.6	82.7	55	0.1	6.8	5
5.92	7.2	66.0	50	0.1	8.0	6
5.02	6.0	83.3	100	0.1	10.0	7
6.9	7.15	86.7	50	0.1	16.0	8
5.47	5.9	80.0	30	0.1	4.0	9
8.1	8.1	74.1	40	0.05	5.0	10
5.1	5.6	71.4	80	0.1	15.0	11
7	8.2	71.9	70	0.1	7.3	12

“References

1. Yang, B., Guo, D., Lin, P., Zhou, L., Li, J., Fang, G., Wang, J., Jin, H., Chen, X. A., Wang, S., Hydroxylated Multi-Walled Carbon Nanotubes Covalently Modified with Tris(hydroxypropyl) Phosphine as a Functional Interlayer for Advanced Lithium–Sulfur Batteries. *Angew. Chem. Int. Ed.* **2022**, 61, 2204327.
2. Li, Y., Gao, T., Ni, D., Zhou, Y., Yousaf, M., Guo, Z., Zhou, J., Zhou, P., Wang, Q., Guo, S. J., Two Birds With One Stone: Interfacial Engineering of Multifunctional Janus Separator for Lithium-Sulfur Batteries. *Adv. Mater.* **2022**, 34, 2107638.
3. Zhang D., Wang S., Hu R. M., Gu J. A., Cui Y. L. S., Li B., Chen W. H., Liu C. T., Shang J. X., Yang S. B., Catalytic Conversion of Polysulfides on Single Atom Zinc Implanted MXene toward High-Rate Lithium-Sulfur Batteries. *Adv. Funct. Mater.* **2020**, 30, 2002471.
4. Su, L., Zhang, J., Chen, Y., Yang, W., Wang, J., Ma, Z., Shao, G., Wang, G., Cobalt-Embedded Hierarchically-Porous Hollow Carbon Microspheres as Multifunctional Confined Reactors for High-Loading Li-S Batteries. *Nano Energy* **2021**, 85, 105981.

5. Zhao, M., Chen, X., Li, X. Y., Li, B. Q., Huang, J. Q., An Organodiselenide Comediator to Facilitate Sulfur Redox Kinetics in Lithium–Sulfur Batteries. *Adv. Mater.* **2021**, 33, 2007298.
6. Liu, W., Lei, M., Zhou, X., Li, C., Heterojunction Interlocked Catalysis-Conduction Network in Monolithic Porous-Pipe Scaffold for Endurable Li–S Batteries. *Energy Storage Mater.* **2023**, **58**, 74.
7. Ma, C., Zhang, Y., Feng, Y., Wang, N., Zhou, L., Liang, C., Chen, L., Lai, Y., Ji, X., Yan, C., Wei, W. Engineering Fe–N Coordination Structures for Fast Redox Conversion in Lithium–Sulfur Batteries. *Adv. Mater.* **2021**, 33, 2100171.
8. Huang, T., Sun, Y., Wu, J., Jin, J., Wei, C., Shi, Z., Wang, M., Cai, J., An, X. T., Wang, P., Su, C., Li, Y. Y., Sun, J., A Dual-Functional Fibrous Skeleton Implanted with Single-Atomic Co–N_x Dispersions for Longevous Li–S Full Batteries. *ACS Nano* **2021**, 15, 14105.
9. Wang, M., Sun, Z., Ci, H., Shi, Z., Shen, L., Wei, C., Ding, Y., Yang, X., Sun, J., Identifying the Evolution of Selenium-Vacancy-Modulated MoSe₂ Precatalyst in Lithium–Sulfur Chemistry. *Angew. Chem. Int. Ed.* **2021**, 60, 24558.
10. Yu, S., Sun, Y., Song, L., Cao, X., Chen, L., An, X., Liu, X., Cai, W., Yao, T., Song, Y., Zhang, W., Vanadium Atom Modulated Electrocatalyst for Accelerated Li–S Chemistry. *Nano Energy* **2021**, 89, 106414.
11. Zhou, G., Zhao, S., Wang, T., Yang, S. Z., Johannessen, B., Chen, H., Liu, C., Ye, Y., Wu, Y., Peng, Y., Liu, C., Jiang, S. P., Zhang, Q., Cui, Y. Theoretical Calculation Guided Design of Single-Atom Catalysts toward Fast Kinetic and Long-Life Li–S Batteries. *Nano Lett.* **2020**, 20, 1252.
12. Fang, D., Sun, P., Huang, S., Shang, Y., Li, X., Yan, D., Lim, Y. V., Su, C. Y., Su, B. J., Juang, J.-Y., Yang, H. Y. An Exfoliation–Evaporation Strategy to Regulate N Coordination Number of Co Single-Atom Catalysts for High-Performance Lithium–Sulfur Batteries. *ACS Mater. Lett.* **2021**, 4, 1.”

5. Chlorine bridge bond-enabled binuclear copper complex is reported for electrocatalyzing lithium-sulfur battery reaction. Since the electrocatalysis reaction is highlighted in the topic, it is suggested to make a discussion on the electrocatalysis reaction by explaining the reaction intermediate and its active energy toward (or compared to) the lithium-sulfur reaction.

[Suggestion] Please give more information on the electrocatalysis reaction.

Our response:

We thank the reviewer for the instructive comment. Following the reviewer’s suggestion, we have tested the active energy (E_a) for each conversion reaction step of sulfur intermediates to further reflect the catalytic activity of Cu-2 according to the reported testing approach (*Nat. Energy* **2023**, 8, 84; *Nat. Catal.* **2020**, 3, 762; *Nat. Commun.* **2022**, 13, 202). The EIS tests under different temperatures were performed at various voltages where critical sulfur reactions occur (Figure S21). Specially, Nyquist plots at 2.4 and 2.1 V respectively represent the conversion step of soluble LiPSs, and Li₂S₄ to Li₂S₂/Li₂S. After fitting the circuits, Arrhenius equation was applied to

calculate the values of E_a (Figure S22). The related discussions have been also introduced into our revised manuscript.

“The activation energy (E_a) for each conversion reaction step of sulfur intermediates to further reflect the catalytic activity of Cu-2. Electrochemical impedance spectra (EIS) tests under different temperatures were performed at various voltages where critical sulfur reactions occur (Figure S21). Particularly, Nyquist plots at 2.4 and 2.1 V respectively represent the conversion step of soluble LiPSs, and Li_2S_4 to $\text{Li}_2\text{S}_2/\text{Li}_2\text{S}$. After fitting the circuits, Arrhenius equation was applied to calculate the values of E_a (Figure S22). As displayed in Figure S22d, the S/Cu-2 cathode obtains the lowest the values of E_a for each voltage among the three samples, further corroborating the high catalytic activity of Cu-2 for the whole sulfur conversion reaction.”

Figure S21. EIS curves of the S/Cu-2 (a–d), S/Cu-1(e–h) and S/rphenGO (i–l) at different voltages under various temperatures.

Figure S22. Arrhenius plots of the S/Cu-2 (a), S/Cu-1(b) and S/rphenGO (c) at various voltages; (d) activation energies of different cathodes at various voltages.

REVIEWERS' COMMENTS

Reviewer #1 (Remarks to the Author):

The authors have addressed the reviewers' comments satisfactorily and I believe that the manuscript is ready for publication.

Reviewer #2 (Remarks to the Author):

The authors answered all my questions. I am satisfied with the answers and revisions. Since the authors have improved the quality of this article in the light of the reviewer's comments, this article can be accepted for publication.

REVIEWERS' COMMENTS

Reviewer #1 (Remarks to the Author):

The authors have addressed the reviewers' comments satisfactorily and I believe that the manuscript is ready for publication.

Our response:

We are grateful for the positive comment to make our manuscript meet the high standard of the journal.

Reviewer #2 (Remarks to the Author):

The authors answered all my questions. I am satisfied with the answers and revisions. Since the authors have improved the quality of this article in the light of the reviewer's comments, this article can be accepted for publication.

Our response:

Many thanks for the instructive comment from the reviewer to make great improvements to our manuscript.